



# The effect of GCM biases on global runoff simulations of a land surface model

Lamprini V. Papadimitriou[1], Aristeidis G. Koutroulis[1], Manolis.G. Grillakis[1], Ioannis.K. Tsanis[1,2]

[1]Technical University of Crete, School of Environmental Engineering, Chania, Greece
[2]McMaster University, Department of Civil Engineering, Hamilton, ON, Canada

*Correspondence to*: I. K. Tsanis (tsanis@hydromech.gr)

## Abstract

Global Climate Model (GCM) outputs feature systematic biases that render them unsuitable for direct use by
impact models, especially for hydrological studies. To deal with this issue many bias correction techniques
have been developed to adjust the modelled variables against observations, focusing mainly on precipitation
and temperature. However most state-of-art hydrological models require more forcing variables, additionally
to precipitation and temperature, such as radiation, humidity, air pressure and wind speed. The biases in these
additional variables can hinder hydrological simulations, but the effect of the bias of each variable is
unexplored. Here we examine the effect of GCM biases on historical runoff simulations for each forcing
variable individually, using the land surface model JULES set up at the global scale. Based on the quantified
effect, we assess which variables should be included in bias correction procedures. To this end, a partial
correction bias assessment experiment is conducted, to test the effect of the biases of six climate variables
from a set of three GCMs. The effect of the bias of each climate variable individually is quantified by
comparing the changes in simulated runoff that correspond to the bias of each tested variable. A methodology
for the classification of the effect of biases in four Effect Categories (ECs), based on the magnitude and
sensitivity of runoff changes, is developed and applied. Our results show that, while globally the largest
changes in modelled runoff are caused by precipitation and temperature biases, there are regions where runoff
is substantially affected by and/or more sensitive to radiation and humidity. Global maps of bias ECs reveal
the regions mostly affected by the bias of each variable. Based on our findings, for global scale applications,
bias correction of radiation and humidity, in addition to that of precipitation and temperature, is advised.
Finer spatial scale information is also provided, to suggest bias correction of variables beyond precipitation
and temperature for regional studies.





## 1 Introduction

In recent years, there is a strong consensus on the changes in climate caused by increased concentrations of anthropogenic green-house gas emissions (King et al., 2015; O'Neill et al., 2017; IPCC, 2013). Under the pressing circumstances of a warming world, scientific research has focused on estimating the range of

changes in the future climate and the effectiveness of different adaptation strategies. The main tool for the investigation of future climate is the utilization of Global Climate Models (GCMs). GCMs are based on physical principles that describe the components of the climate system, such as cloud formation and water and energy flux exchanges.

Although each generation of GCMs shows improvements compared to its predecessor (Koutroulis et al.,

2016), climate model outputs still contain substantial biases that express as deviations of the modelled climate variables from respective historical observations. These inherent biases can emanate from misrepresentations of physical atmospheric processes (Maraun, 2012), from uncertainties regarding the boundary and initial model conditions (Bromwich et al., 2013), and from the relatively coarse resolution employed by the GCMs (Katzav and Parker, 2015). As a result, outcomes of hydrological climate change impact studies have been

reported to become unrealistic without a prior adjustment of climate forcing biases (Ehret et al., 2012; Hansen et al., 2006; Harding et al., 2014; Sharma et al., 2007). To overcome this limitation, various bias correction techniques have been developed to post-process climate model data to statistically match observations. Bias correction methods are calibrated based on a historical time period for which observations are available. The adjustment is then applied to both modelled historical period and to the period beyond the time-frame of the

observations.

Bias correction procedures have mainly focused on adjusting the biases of precipitation and/or temperature (Christensen et al., 2008; Li et al., 2010; Miao et al., 2016; Photiadou et al., 2016; Piani et al., 2010). These variables have traditionally been prioritized for bias correction as they are considered as the most important driving variables of hydrological processes in modelling applications -even though from a physical

perspective radiation is the driving force of the hydrological cycle. However, many state-of-the-art regional and global Hydrological Models (GHMs) and Land Surface Models (LSMs) require -apart from precipitation and temperature- additional meteorological forcing, such as solar radiation, air humidity, surface air pressure and wind speed (a summary of the input variables needed by various hydrological models can be found in the Supplement of Hattermann et al. (2016)). For this reason, biases in variables like radiation, humidity and

wind speed can hinder the representation of hydrological fluxes such as runoff, evapotranspiration (ET), snow accumulation and snowmelt by the impact models (Hagemann et al. 2011; Haddeland et al. 2012), indicating that bias correction should be extended to include more input variables.





Bias correction itself also has limitations, as it is a demanding process, both in terms of computational cost and of the involved methodological development. Moreover, the use of bias correction is challenged by conceptual pitfalls such as the disruption of the physical consistency of climate variables, the mass/energy balance and the omission of correction feedback mechanisms to other climate variables (Ehret et al., 2012).

For these reasons, it is worth examining whether the effect of biases of input variables on hydrological outputs justifies the use of bias correction. Even though this information would be key for making informed decisions on the variables that should be bias corrected for a specific model application, few relevant studies can be found in literature. Some insight is given by Haddeland et al. (2012), who investigates the combined effect of bias correcting radiation, humidity and wind speed in addition to precipitation and temperature on

hydrological simulations. However, the extent to which individual forcing variable biases affect hydrological simulations and the way that this effect varies spatially are important research questions that remain open.

Here we investigate the effect of the biases in GCM climate variables on the historical runoff output of a large scale LSM. To this end, we firstly quantify the improvements in the representation of historical modelled runoff when bias corrected variables are used as forcing. Secondly, we examine the individual

effect that the bias of each climate variable can have on runoff simulations. This way we can provide an assessment of the variables beyond precipitation and temperature that may be considered as "priority" variables for bias correction, due to their possible pronounced effect on hydrological simulations.

## 2    Methods

### 2.1    The JULES land surface model

Hydrological simulations were performed with the Joint UK Land Environment Simulator (JULES) model (Best et al., 2011). JULES is a physically based model that calculates water, energy and carbon exchanges between the land surface and the atmosphere. The science modules that comprise the model are: surface energy fluxes, snow cover and surface hydrology, soil moisture and temperature, soil carbon, vegetation dynamics and plant physiology. The model requires seven climate variables as forcing, namely: precipitation,

temperature, longwave and shortwave radiation, specific humidity, surface pressure and wind speed. Runoff production in JULES has two components. The first one is surface runoff, produced by the infiltration excess mechanism. The second one is subsurface runoff (or drainage from the bottom of the soil column), which is calculated as a Darcian flux under the assumption of zero gradient of matric potential. Calculation of potential evaporation follows the Penman–Monteith approach (Penman, 1948). Water held at the plant canopy

evaporates at the potential rate while restrictions of canopy resistance and soil moisture are applied for the simulation of evaporation from soil and plant transpiration from potential evaporation (Best et al., 2011). For a detailed description of JULES the reader can refer to the model description papers of Best et al. (2011) and



Clark et al. (2011). Examples of recent model applications on climate change impact assessments can be found in the studies of Papadimitriou et al. (2016), where JULES is used to investigate future water availability in Europe, and Grillakis et al. (2016), who estimated the climate induced changes in soil temperature regimes.

*2.2 Model setup and outputs*

JULES was run at the global scale, with a spatial resolution of 0.5 degrees. A daily time step was employed for all the model runs. To warm up the model, 10 spin-up cycles from 1973 to 1978 were performed before each main run. The main runs span from 1978 to 2010 but only the time period of 1981 to 2010 is used for the analysis. The model outputs are produced with a daily time resolution.

*2.2  Hydrological evaluation*

This study focuses on the runoff production output of JULES, hereafter denoted as RF. For the assessment of model performance, RF is aggregated at the basin level to allow for comparison with discharge observations. To this end, RF is converted to discharge at the basin outlet (noted as Q) through a delay algorithm proposed by Zulkafli et al. (2013) and the use of TRIP river routing scheme (Oki and Sud, 1998)

to determine the grid boxes upstream of the basin's outlet.

For the evaluation of JULES' hydrological performance, three metrics are used: Nash-Sutcliffe efficiency (NSE), Percent bias (PBIAS) and the coefficient of determination ($R^2$). The formulas for the calculation of NSE and PBIAS are given in equations 1 and 2:

$$NSE = 1 - \left[\frac{\Sigma(Q_{sim}-Q_{obs})^2}{\Sigma(Q_{obs}-Q_{mean})^2}\right] \qquad (1)$$

$$PBIAS = \left[\frac{\Sigma(Q_{sim}-Q_{obs})*100}{\Sigma Q_{obs}}\right]\% \qquad (2)$$

where $Q_{sim}$ is simulated discharge, $Q_{obs}$ is observed discharge and $Q_{mean}$ is the mean of observed discharge data. Discharge observations were obtained from the Global Runoff Data Centre (GRDC) database for 9 large-scale basins shown in Figure 1.

The evaluation metrics are calculated from monthly discharge data. These are the monthly averages of daily discharge for simulations while observations were obtained in monthly time-steps. Model evaluation was based on the historical period from 1981 to 2010. The months missing from the observed discharge time series were neglected from the calculation of the evaluation metrics.





### 2.3 Climate data

The climate dataset used for bias correction of the GCM data and as a baseline for comparison of the results is the WATCH Forcing Data methodology applied to ERA-Interim data (WFDEI; Weedon et al. 2014). WFDEI data span from 1979 to 2012, but here only the time period from 1981 to 2010 was used. The WFDEI
dataset is based on its predecessor WFD (WATCH Forcing Data; Weedon et al. 2010), which was derived from the ERA-40 reanalysis product (Uppala et al., 2005) after interpolation to half-degree resolution, elevation adjustments and monthly-scale corrections made against gridded station observations of the Climate Research Unit. For detailed information on the derivation of the WFDEI dataset the reader is referred to Weedon et al. (2014).

Data from three GCMs participating in the fifth phase of the Coupled Model Intercomparison Project (CMIP5; Taylor et al. 2012) were used as forcing. Information on the ensemble members can be found in Table 1. Climate model outputs were interpolated to the 0.5º spatial resolution of the WFDEI dataset, using the nearest-neighbor method.

### 2.4 Bias correction method

The bias correction methodology presented by Grillakis et al. (2013), namely Multi-segment Statistical Bias Correction (MSBC), is used to adjust the biases in precipitation. The methodology has already been used in the Bias Correction Intercomparison Project (BCIP) (Nikulin et al., 2015) and in a number of climate change impact studies (Grillakis et al., 2016; Papadimitriou et al., 2016). MSBC follows the principles of quantile mapping correction techniques and was originally designed and tested for GCM precipitation adjustment.

According to the method, the Cumulative Distribution Function (CDF) space is split into discrete segments and then the individual quantile mapping correction is applied on each segment, achieving better fit of the parametric equations on the data and thus better correction, especially on the CDF edges. The optimal number of segments is estimated by the Schwarz Bayesian information criterion to balance between complexity and performance. A modification of the methodology is used for bias adjustment of the rest of the variables that

were used. Here the methodology is modified to use linear functions instead of the gamma that were used in the original methodology. This change allows for the facilitation of negative variable values that the gamma functions cannot simulate. Hence the methodology becomes more universal to be used in different variable types and distributions. An additional methodological change is performed to the edge segments correction, which are explicitly corrected using only the difference between the historical period model data and the

observations. This provides rigidity to the correction, avoiding unrealistic temperature values at the edges of the corrected data CDF. This choice costs to the methodology the persistence of a small portion of the bias in the corrected data. As MSBC methodology belongs to the parametric quantile mapping techniques, it





shares their advantages and drawbacks. A comprehensive analysis of advantages and disadvantages of the methods that follow the quantile mapping comparing to others can be found in Maraun et al. (2010) and Themeßl et al. (2012).

### 2.5  Experimental design

In order to examine the effect of each forcing variable's bias on runoff we designed and implemented an experiment comprised of two parts (bias assessment and partial correction bias assessment) and nine sets of JULES' runs in total. A graphical description of the performed experiment is shown in Figure 2. Climate data from 3 GCMs and the WFDEI dataset are used as JULES' forcing. The sets of runs forced with GCM data, include three model runs –one per GCM. Then the analysis progresses using the ensemble mean.  The time

span of this analysis is the historical period 1981-2010. This is also the time span of the period used for bias correction of the GCM output.

### 2.6  Bias assessment

The first part of the experiment is to assess initial and remaining biases in the forcing data and in simulated runoff. Initial bias refers to the difference between raw GCM variables and the respective WFDEI variables.

Remaining bias is the bias in the forcing variables after the bias correction, i.e. the difference between bias corrected GCM variables and the respective WFDEI variables. Referring to runoff, "initial" and "remaining" biases are defined as the difference between runoff simulations forced with raw and bias corrected forcing respectively from simulations forced with the WFDEI dataset. This definition is employed to shorten and simplify the expressions used in this paper (i.e. "initial bias in runoff" instead of "the difference between

runoff forced with raw GCM data and WFDEI data"). In this part of the experiment, three sets of JULES' runs were conducted:

i)       forced with WFDEI (WFDEI)

ii)      forced with uncorrected climate data (Raw)

iii)     forced with bias corrected climate data (BC).

### 2.7  Partial correction bias assessment

For the second part of the experiment -the partial correction bias assessment- six more sets of JULES' runs were performed. In each of these runs, one of the six forcing variables (precipitation, temperature, radiation, humidity, surface pressure and wind speed) is used in its raw form while the rest of the input forcing is bias corrected. The partial correction assessment runs are symbolized as NobcV (NOt Bias Corrected variable V),

where V is one of the six forcing variables: precipitation (P), temperature (T), radiation (R), specific humidity (H), surface pressure (Ps) and wind (W). It has to be noted here that -longwave (Rl) and shortwave (Rs) were





examined together, hence in respective NobcR run, both shortwave and longwave radiation were forced in uncorrected form. Partial correction assessment is composed as a tool to quantify the individual effect of each forcing variable on runoff but is not designed to suggest and assess run formats.

The simulated runoff of each partially corrected input is compared to the respective simulation in which all
input variables are bias corrected (denoted as BC). This comparison allows us to assess the "loss" of the performance of simulations when a variable is neglected from the bias correction procedure. It must be noted however that the "loss of performance" concept bears the assumption that the BC simulation is closer to the WFDEI simulation comparing to a partially corrected set.

*2.8    Categorization of individual variable bias effects*

A new framework for the classification of the effects of forcing variables' biases on modelled runoff is developed and implemented. The classification employs the comparison of the bias in each forcing variable ($\Delta V$) and the corresponding relative effect in simulated runoff ($\Delta RF$), discretizing four different categories (Figure 3). To facilitate the comparison among the different forcing variables, $\Delta V$ and $\Delta RF$ are expressed as percentages. More specifically, $\Delta V$ and $\Delta RF$ are defined as follows.

$\Delta V$ is the difference between the raw and the bias corrected variable value, divided by the bias corrected variable value. $\Delta V$ is estimated by equation Eq. 3.

$$\Delta V = \frac{Raw\ variable - BC\ variable}{BC\ variable} * 100\% \quad (3)$$

As an exception, for temperature $\Delta V$ refers to the absolute difference between raw and bias corrected temperature (in K).

$\Delta RF$ expresses the effect of a variable's bias on runoff and is calculated from the difference between runoff forced with all bias corrected variables except for the examined variable V (NobcV) and runoff forced with all bias corrected variables (BC), divided by the runoff of all bias corrected variables (BC). $\Delta RF$ is estimated by equation Eq. 4.

$$\Delta RF = \frac{RF\ from\ NobcV - RF\ from\ BC}{RF\ from\ BC} * 100\% \quad (4)$$

Sensitivity of runoff to changes in forcing variables (S) is the fraction of runoff change over the forcing variable change and serves as a measure to assess the relative magnitude of $\Delta RF$ compared to $\Delta V$. When $\Delta RF$ is sensitive to $\Delta V$, relatively smaller changes in the variable should cause relatively larger changes in runoff and vice versa. Sensitivity is in general dimensionless, but for temperature has units of $K^{-1}$. S is estimated by:





S=ΔRF/ΔV  (5)

In total, there are 6 sets of ΔVs and 6 sets of ΔRFs, one for each examined variable and experiment respectively, and 6 sets of sensitivities (S). The absolute values of ΔV, ΔRF and S denoted as |ΔV|, |ΔRF| and |S| are used to avoid dealing with the sign of the changes and rather focus on their magnitude.

As shown in Figure 3, the effect of each variable's bias (|ΔV|) on runoff (|ΔRF|) is separated into four different categories according to two rules. The first rule is the characterization of |ΔRF| among all the experiments (except ΔT) as "low" or "high" relatively to its median value, shaping the ordinate y=median(|ΔRF|). The second rule is the characterization of sensitivity |S| as high or low relatively to its median value. The later forms a bisectrix s=median(|S|). These two rules form the four categories of Figure 3. In the case of

temperature, median(|S|) was explicitly recalculated. Combinations of the two rules result to four different "Effect Categories" (ECs) presented in decreasing order of the effect of a variable's bias on runoff:

I.   High change and high sensitivity (ECI)

II.  High change and low sensitivity (ECII)

III. Low change and high sensitivity (ECIII)

IV. Low change and low sensitivity (ECIV)

*2.9    Regional scale bias assessment*

Regional focus is given at 24 regions and 9 hydrological basins. The regions were selected from the 26 regions presented in Giorgi & Bi (2005) (in our study Alaska and Greenland are excluded from the analysis). The hydrological basins were selected to cover different hydro-climatic regimes, in conjunction with GRDC

data availability. The selected regions and basins are shown in Figure 1. The abbreviations of the regions' names can be found in Table 2.

**3    Results**

*3.1    Long-term annual biases in forcing variables at the global scale*

Global maps of the initial and remaining biases of the forcing variables are shown in Figure 4. In general

terms the remaining biases are smaller than the initial ones by one to two orders of magnitude. For precipitation (Figure 4a), the largest initial wet biases are observed for regions with high mountain ranges (the Andes in South America, the Alaska Range and the Rocky Mountains in North America and the Himalayas in Asia) and for the tropical African and Indonesian regions. The most dominant initial dry biases are found in the Amazon region. Only a very small percentage (0.75%) of the land surface has small biases

(-0.01 to 0.01 mm/day) while the largest biases (>5 mm/day or <-5 mm/day) occupy 31.18% of the land





surface. The remaining biases in precipitation are small (up to 0.01 mm/day in absolute terms, for 80.32% of the land surface) and located in the tropics. The initial biases in temperature are cold biases for 57.82% of the land surface while warm biases (mainly found in the Alaskan, Greenland, north and central Asia regions as well as in the Mediterranean and the Andes) occupy 42.12% of the land surface (Figure 4b). Initial biases

greater than 2 K in absolute terms cover approximately one third of the land surface (34.74%). After bias adjustment, the remaining temperature bias is less than 0.1 K for the vast majority of the land surface (97.27%).

The initial biases of longwave and shortwave radiation (Figure 4c and Figure 4d respectively) exhibit similar spatial variations but have different signs. Shortwave radiation shows a greater extent of large biases (>50

W/m$^2$ in absolute terms) compared to longwave radiation (8.16% as opposed to 2.95% of the land surface). Initial biases in specific humidity are greater than $10^{-3}$ kg/kg (1g/kg), in absolute terms, for one quarter of the land surface (23.65%) (Figure 4e). The largest biases in surface pressure (>50 or <-50 HPa) occupy 10.01% of the land surface and are found in the areas where high mountain ranges are located (Rocky Mountains, Andes, Himalayas) (Figure 4f). The remaining bias in surface pressure is less than 0.1 HPa (in absolute terms)

for most of the land surface (96.50%). For more than half of the land surface (55.79%) wind's initial biases are larger than 0.5 m/s or smaller than -0.5 m/s (Figure 4g). The remaining biases of the wind variable range between -0.01 and 0.01 m/s for the majority of the land surface (87.71%).

Generally, the initial GCM biases in precipitation and temperature are more pronounced over high mountainous regions and the tropics. Recent studies argue towards a dependency between biases and altitude.

According to the study of Haslinger et al. (2013), both temperature and precipitation biases of a GCM tested over the Alpine Region, show increasing trends with height. Regarding the tropics, various studies show increased GCM biases in these regions compared to model performance in other climate zones (Koutroulis et al., 2016; Randall et al., 2007; Solman et al., 2013). The initial surface pressure biases are also linked to altitude, as surface pressure heavily depends on elevation. Initial biases in surface pressure have an elevation-

similar pattern and could be a result of the different spatial resolution of the elevation model in the GCMs and WFDEI. The WFDEI dataset resolution is 0.5 degrees while the original GCM spatial resolution is considerably lower (around 2.5 degrees). GCM surface pressure is simulated taking into account a relatively low resolution elevation model. Although GCM surface pressure is interpolated to the WFDEI resolution, this does not correct the elevation induced error in the GCM simulations.

The remaining biases in precipitation at the tropical regions were also identified and discussed extensively by Grillakis et al. (2013) and are related to the error in the CDF approximation during bias correction. For the rest of the variables, the remaining bias although not actually zero is very close to zero (well below the



smallest positive and above the smallest negative rank in the legend, e.g. below -0.1 K and below 0.1 K for temperature). The color scale in Figure 4 was selected with the intention of showing the remaining biases, but this does not mean that their values are accountable. They are rather trace errors occurring due to truncation numerical errors during the bias correction process. Hence the remaining biases (except for

precipitation) could not be attributed to a specific mechanism.

### 3.2  *Regional and seasonal biases in forcing variables*

Figure 5 illustrates the initial biases of the GCM ensemble, spatially aggregated over 24 regions of the globe. To account for possible seasonality variations, the biases are calculated for the annual mean (ANN) and for the December-January-February (DJF) and June-July-August (JJA) means. The remaining biases are not

shown because their regionally aggregated values are negligible and would be indistinguishable on the Figure.

Precipitation biases are less pronounced in Europe (NEU, MED, NEE) and in central and north Asian regions (CAS, NAS). The wettest precipitation biases are encountered in the equatorial and Southern Africa (EQF, SQF and SAF) and concern DJF precipitation (Figure 5). The driest biases are found for the CAM, AMZ and

SAS regions, for JJA precipitation. Temperature displays cold biases in most regions. A notable exception is the warm bias in DJF temperature in the NAS region, which is the most pronounced temperature bias found. Generally the DJF temperature biases are the largest, followed by ANN, while the JJA season has the smallest temperature biases.

The two radiation components, long-wave (Rl) and short-wave (Rs) radiation, show an inverse behavior in

their biases (Figure 5). That is to say, in regions where Rl has negative biases Rs exhibits positive biases and vice versa. According to Demory et al. (2014), overestimation of shortwave radiation is a common issue amongst the GCMs. Negative biases are dominant for Rl in contrast to the Rs variable, which mostly shows positive biases. Specific humidity has negative biases over the north part of the African continent (SAH, WAF, EAF, EQF), central and south America (CAM, AMZ, CSA) and south Asia (SAS). Positive humidity

biases are identified in the south part of Africa (SQF and SAF) and north America (WNA, CNA and ENA).

Surface pressure shows almost exclusively positive biases (Figure 5). The regions that distinguish for the largest biases are MED, SEA, SAH, SAF, CAM, CSA and SSA. The most dominant negative wind speed bias is found in NAU. Most of the African continent (SAH, WAF, EAF, EQF, SQF) and of South America (AMZ, CSA) also have negative biases in wind. The largest positive biases are encountered in the southern

part of South America (SSA) for the JJA season and for the DJF season in regions of North America (WNA, CAM), Europe (MED) and Asia (CAS, TIB, SEA).



Hydrology and
### 3.3  Model evaluation

In order to assess JULES' performance, we compare discharge modelled with WFDEI and with the raw GCM dataset to discharge observations for nine study basins. Figure 6 shows the seasonality of observed and modelled discharge and a comparison of the evaluation metrics of the two sets of simulations (WFDEI and raw GCM) are presented in Figure 7.

For seven out of the nine basins (Amazon, Congo, Volga, Ganges, Danube, Elbe and Kemijoki) seasonality is well captured by the WFDEI simulation (Figure 6). In contrast, the raw GCM simulation exhibits significant positive and negative biases for these seven basins. For the two remaining basins however (Mississippi and Lena), seasonality is better captured by the raw GCM simulation. The WFDEI run results in positive NSE values (0.24 to 0.94) for all the basins. On the contrary the raw GCM run results in negative NSE values for six out of the nine basins. PBIAS indicates that the raw GCM simulation exhibits greater deviations from observations than the WFDEI run for most basins (exceptions are Mississippi, Lena, Ganges and Danube). Finally, the $R^2$ metric shows that the linear correlation between simulations and observations is stronger for the WFDEI run for seven out of the nine basins (exceptions are Mississippi and Elbe). For both simulations the lowest $R^2$ value is reported for the Congo basin (0.45 and 0.2 for the WFDEI and raw GCM runs respectively). The best correlations per simulation are found for Ganges for the WFDEI run (0.99) and for Amazon for the raw GCM run (0.94).

The shown persistent departure from the mean climatology of discharge could include three types of errors. The first is the error stemming from the insufficient description of the runoff processes by the land surface model and from the routing algorithm. The second type of error is a result of errors in the forcing datasets (either observational or GCM output) with regards to depicting the real climatic drivers. A third possible error comes from the comparison of naturalized discharge of the simulations with measured discharge due to influences like abstractions and dams regulating the natural river flow. An extra error component, which is not considered here, could result from errors in discharge measurements. We believe that decreased model performance in some basins is mainly a result of the first type of error (model deficiencies), which are enhanced or masked by the errors in the forcing datasets.

The model evaluation has revealed two basins (Mississippi and Lena) for which raw GCM forced discharge simulations outperform the WFDEI simulations. For Mississippi, the WFDEI run gives higher discharge than the observations throughout the year, revealing a deficiency of the model to capture the water balance of this basin. Most of the Mississippi extent is in the CNA region, where negative precipitation biases have been documented (Figure 5). Thus, the raw GCM run is forced with less precipitation compared to WFDEI and less discharge is produced, masking the model deficiency in this basin and improving the metrics of model





performance. It is also important to note that the range of the raw GCM simulations is quite broad especially for a three member ensemble. The upper range of the GCM ensemble exceeds the WFDEI simulated runoff during almost half the seasonal cycle. This indicates that the individual ensemble members would not necessarily outperform the WFDEI run and that, for this specific basin, the ensemble averaging has possibly

produced a "false-positive" in model performance. In this particular basin, model performance may also be hindered due to the comparison of naturalized with actual discharge, as Mississippi is a heavily regulated river. For Lena, the WFDEI run underestimates measured discharge by about 40%. The Lena basin falls into the extent of the NAS region, for which positive precipitation biases have been documented (Figure 5). The extra water in the raw GCM run counteracts the tendency of the model to underestimate discharge in the

Lena basin, resulting in an improved model performance. In the context of the present study we are not able to identify the exact reasons why model performance is hindered in some basins. It is unrealistic for a global LSM to achieve top performance around the world, even if the tedious work of its calibration is practiced (Hattermann et al., 2016). Due to its global nature, some fixes in some regions could result in deteriorations in performance in other parts of the land surface. Thus, the interpretation of the following analysis of the

present study should consider the model deficiencies revealed at this section.

*3.4    Long-term biases in runoff at the global scale*

Figure 8 shows the initial and remaining biases in runoff, derived from ANN, DJF and JJA long term means. As with the biases in the input forcing variables, the remaining bias in runoff is one to two orders of magnitude smaller than the initial bias. Hence, the use of bias corrected data led to an improved representation

of runoff by the model, compared to the baseline of the WFDEI run. Accordingly, the studies of Teutschbein & Seibert (2012) and Rojas et al. (2011) found that hydrological simulations are substantially improved with the use of bias corrected forcing.

Regarding the raw GCM run, the largest runoff underestimation biases (<-5 mm/day) are encountered in central-north America, the central-east part of South America and East Asia. The most pronounced runoff

overestimation biases are found in the west part of North and South America, in equatorial, south Africa, northern Europe, the Tibetan region and Indonesia. Initial runoff biases are larger than 1mm/day in absolute terms for 16.26%, 14.85% and 20.18% of the land surface respectively for ANN, DJF and JJA. The differences between the seasonal means (DJF, JJA) and the annual mean (ANN) are in general subtle. Yet, the increases in runoff overestimation biases in DJF in south equatorial Africa and in JJA in the Tibetan

plateau are worth noting. Large initial biases (>5mm/day in absolute terms) in seasonal means occupy a greater percentage of the land surface compared to annual mean (0.70% for ANN, compared to 1.25% and 1.97% for DJF and JJA respectively).




The remaining biases in runoff range from -0.1 to 0.1 mm/day for the majority of the land surface (95.19%, 87.40% and 80.30% for ANN, DJF and JJA respectively). Negligible biases (smaller than 0.01 mm/day in absolute terms) are found for more than one third of the land surface (specifically for 38.06% of the land area for ANN, 37.60% for DJF and 34.42% for JJA). The (negative) remaining bias in ANN runoff is more

pronounced in the west Amazonian region. This probably corresponds to the remaining bias in precipitation identified for the Amazon region (Figure 4). In addition to the significant reduction of the biases in runoff forced with bias corrected data, it can be observed that the remaining biases have switched signs compared to the initial biases. This means that in regions where the initial bias in runoff is positive (negative), thus the raw GCM forced runoff is larger (smaller) than runoff forced with WFDEI, the use of bias corrected forcing

results in runoff slightly lower (higher) than WFDEI runoff. A respective behavior was not observed in the initial and remaining biases of the most impacting forcing variables (P and T) but it was, to an extent, present for other variables (Rl, Rs and H). Thus, the "overcorrection" manifested for bias corrected runoff compared to WFDEI runoff cannot be attributed to remaining biases in precipitation and temperature. Instead, it could plausibly be associated with the compound effect of the remaining biases in part of (or in all other) forcing

variables.

### 3.5  Effect of each forcing variable's bias on runoff

The effect that the bias of each forcing variable can have on runoff is investigated here, by comparing runoff from the bias corrected run to the partial correction assessment runs. The results are shown in Figure 9, for ANN, DJF and JJA averages.

First, we discuss the runoff differences calculated from the ANN period. Precipitation and temperature are the only two variables that cause runoff differences larger than 5 mm/day (in absolute terms) when neglected from bias correction. However, these differences regard a very small percentage of the land surface: 0.61% for precipitation and only 0.02% for temperature. Moreover, precipitation bias causes changes in runoff greater than 1 mm/day (in absolute terms) for 14.28% of the land area. Such changes for the other variables

occupy a significantly smaller fraction of the land area (ranging from 1.21% for temperature to 0.05% for wind). Based on the above it can be stated that precipitation is the variable that mostly affects runoff response. Precipitation bias causes both wet and dry biases in different regions of the land surface, with a pattern that closely resembles the effect of the initial GCMs' biases on runoff (Figure 8). A similar pattern between precipitation and runoff biases was also observed by Teng et al. (2015), who noted that precipitation errors

are magnified in modelled runoff. Temperature biases result in runoff overestimation for around 60% of the land surface (e.g. over west- and east-North America, the Amazon region, equatorial Africa, northern Europe and parts of Asia) and runoff underestimation for around 40% (example regions: parts of central-south

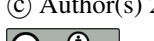



America and of central Asia). Temperature biases correspond with small changes in runoff (up to 0.01 mm/day in absolute terms) over about one third of the land area. Excepting the radiation components from the bias correction procedure produces negative runoff changes for the majority of the land surface (67.60%), while for around 80% of the land surface the differences in runoff range between -0.1 and 0.1 mm/day. The bias in the specific humidity variable corresponds to runoff overestimations for 64% of the land area. The areas of runoff overestimation are mainly located at the higher latitudes (northern part of north America, Europe, north Asia). For 36.43% of the land surface, changes in runoff due to specific humidity biases span between 0.1 and 0.5 in absolute terms. Surface pressure and wind are the variables that show the smaller effect on the hydrological output, as their exclusion from bias correction corresponds to small changes in runoff (less than 0.1 mm/day in absolute terms) for the vast majority of the land surface (around 94% and 92% of the land surface respectively for surface pressure and wind speed). The most pronounced differences in runoff due to surface pressure biases are negative and are encountered over the high mountain ranges' regions of South America and Asia (Andes and Himalayas respectively).

The patterns of runoff changes due to the biases of the forcing variables derived from annual (ANN) and seasonal (DJF, JJA) averages show only subtle variations. In general the above analysis on the ANN runoff differences applies also to the seasonal values, with small variations on the land fractions that show a specific response to forcing biases.

From this analysis it can be deduced that apart from the main hydrological cycle drivers (precipitation and temperature), radiation and specific humidity can also pose a substantial effect on runoff, especially for specific regions. These findings will be further investigated and discussed in the following sections. Other studies also advocate towards the considerable effect that biases in radiation (Mizukami et al., 2014) and humidity (Masaki et al., 2015) can have on hydrological fluxes.

### 3.6   Runoff sensitivities to forcing variables

Sensitivity of runoff changes to the biases of the forcing variables is examined by exploring the relationship between the input forcing biases ($\Delta V$) and the corresponding changes in runoff ($\Delta RF$). The regional variation of this relationship is also investigated. Figure 10 shows scatterplots of $\Delta RF$ versus $\Delta V$ for each examined variable, for 10 selected regions. The dots in each scatterplot correspond to the land grid boxes of each region. The presented regions are selected as representative of different parts of the land surface, as the number of the regions shown in the manuscript had to be reduced for clarity of the results. Scatterplots of the 24 examined regions can be found in the Supplement of this paper. The median values of $\Delta V$, $\Delta RF$ and S of the land grid boxes of each region, for the 24 examined regions, are shown in Table 3.



The correlation between the six ΔVs and respective ΔRFs differs substantially between the examined regions. Generally, the correlations show a non-uniform behavior, identified by the highly scattered data clouds. This implies a high spatial variability of runoff sensitivity to the examined variables.

For precipitation, the ΔRF over ΔP relationship exhibits a nonlinear behavior, indicating that the relative
change is runoff is not proportional to precipitation bias, but also depends on the magnitude of precipitation bias. Renner et al. (2012) also identified nonlinearities in the relationship between relative changes in streamflow and changes in precipitation and argued that nonlinear behavior is a result of the combined effects of water and energy balances. Temperature biases have an inversely proportional and highly nonlinear relationship with changes in runoff. The ΔRF over ΔT relationship is also variant for different regions. For
example, the scatterplots for NEU and WNA indicate that small temperature biases may correspond with large changes in runoff. In contrast, the scatterplot for CAM indicates that larger temperature biases correspond with smaller changes in runoff compared to the other regions. Radiation biases are small but can correspond with high changes in runoff for some regions (WNA, SAS, WAF, AMZ). For specific humidity it can be observed that small positive biases correspond to high changes in runoff for some region (NEU,
MED, WNA and ENA). A different behavior is observed for CAM, SAS, AMZ and CSA where the data cloud is more scattered on the x axis (meaning larger biases in specific humidity) and less scattered on the y axis (i.e. changes in runoff are smaller). Surface pressure has smaller biases compared to the other forcing variables and its effect on runoff also appears reduced. Wind has a wide range of both positive and negative biases which, however, do not seem to affect runoff accordingly.

The variation of the ΔRF over ΔV relationships across the different regions can be attributed to a number of factors. First, it depends on the magnitude and signal of the biases in the forcing variables. As previously shown, these can have significant spatial variations (Figure 4). For example, according to the median values of relative changes in Table 3, some regions are dominated by negative precipitation biases (MED, SAS, AMZ, CSA) and others by positive biases (NEU, WNA, ENA, CAM, WAF, SAU). Second, it reflects the
climatology of each region. The same biases would affect differently regions with different runoff (and evapotranspiration) fractions of each region. The precipitation partitioning to runoff and evapotranspiration is a climate characteristic and is controlled by either water or energy limitations depending on the region. Additionally, we should consider that although we assess the effect of long-term annual biases on long-term annual runoff, the results are still depended on the seasonal cycles of the variables and/or runoff, especially
if the seasonality of precipitation in the region is strong. For example, the same annual bias in temperature would translate differently to runoff changes in a region with precipitation evenly dispersed throughout the year and in another region where most of annual precipitation happens during the summer months. Finally,



as this is a model-based experiment, we should consider whether high sensitivities of some variables for specific regions are a result of over-sensitivity of the model. Vano et al. (2012) documented considerable differences in the spatial distribution of sensitivities to precipitation modelled by five LSMs.

### 3.7 Spatial distribution of bias effect categories

Figure 11 shows global maps of bias effect categories (ECs) for each forcing variable, derived according to the methodology described in Section Categorization of individual variable bias effects. The land area fraction corresponding to each EC is tabulated in Table 4.

Precipitation is the variable whose biases have the largest effect on runoff, with the vast majority of the land surface (92%) corresponding to the high change categories ECI (67.80%) and ECII (24.20%). Radiation has
the second largest land fraction in ECI but temperature has the second largest land fraction in the high change categories (ECI and ECII). Radiation also has the largest land fraction in the high sensitivity categories (ECI and ECIII). As discussed in Section 3.6, this is possibly a result of combining shortwave and longwave radiation for the calculation of the radiation biases. For specific humidity, the most affected areas (ECI) show a significant spatial coherence and are clustered in the higher latitudes of the globe. Surface pressure biases
belong to ECI for around one tenth of the land surface. A dependency between the regions in ECI and terrain elevation can be identified in Figure 11. For wind the majority of the land surface corresponds to ECIV. Still, around one quarter of the land surface belongs to the high change categories (ECI and ECII).

### 4 Discussion

Here we compare our findings to the respective literature to assess the realism of JULES' sensitivity. We use
the median sensitivity value of the grid boxes of each region (Table 3) as the representative sensitivity S for each region. Moreover, we discuss issues of possible model over-sensitivity in particular regions and the caveats of this study.

### 4.1 Sensitivity of runoff to precipitation

Most studies have examined the sensitivity (also reported as elasticity) of runoff (or discharge) to
precipitation. A number of studies have examined sensitivity to precipitation for regions or basins in the United States. Values of runoff sensitivity (S) to precipitation between 1.5 and 2.5 were reported by Sankarasubramanian and Vogel (2003) for the US (WNA, CNA and ENA). Fu et al. (2007) reported values of 1.5 to 1.67 for the Spokane River basin (located in WNA). Vano et al. (2012) found that S to precipitation ranged from 2.2 to 3.3 for different LSMs for the Colorado River basin (also located in WNA). For the
Mississippi River basin (mainly located in CNA), Renner et al. (2012) found that S of streamflow to precipitation is 2.38 and 2.55 using two different methods for sensitivity estimation. For another basin located



in CNA, Brikowski (2015) reported runoff S to precipitation to be 2.64. For the US region, the S values found in this study compare very well with the literature values. Runoff S to precipitation is 2.12 for WNA, 2.54 for CNA and 1.69 for ENA. Many studies report S to precipitation for regions or basins of China. Reported values of runoff S to precipitation in the Yellow River basin (located in EAS) are 1.4 to 1.69 (Fu et al., 2007),

1.6 to 3.9 for 89 catchments of the EAS region (Yang and Yang, 2011), 1.71 and 1.74 (estimates of two different methods) for the headwaters of the Yellow River (Renner et al., 2012). Again, the value found in our study is in good agreement with the literature (S to precipitation for EAS is 1.70).

### 4.2   *Sensitivity of runoff to temperature and other variables*

A number of studies have examined runoff sensitivity to temperature changes. Vano et al. (2012) reported S

to temperature values ranging from -2 to -9 $C^{-1}$ between 5 LSMs for the Colorado River basin (WNA) and Brikowski (2015) reported a value of -0.41 $C^{-1}$ for S to temperature in a basin in CNA. Our values for these regions are substantially lower (-0.13 $K^{-1}$ for WNA and -0.07 $K^{-1}$ for CNA). This divergence could be attributed to two factors. First, to an extent it could be connected to possible non-sensitivities of our model to temperature changes for these regions. Second, the differences could arise from the inclusion (or not) of

the physical link between temperature and other variables in the analysis. Vano et al. (2012) use different LSMs to calculate sensitivities by perturbing daily temperature maxima and minima. These changes also affect the downward longwave radiation and humidity, which are then used by the evapotranspiration routines of the LSMs. In our case, the change in temperature does not interact with radiation and humidity, as those are read as input variables by the model. When temperature is allowed to interact with humidity, increased

temperature will increase the water vapour capacity of the air, and more water will be evaporated. The lack of this physical link in our simulations could, to an extent, explain the decreased sensitivity of runoff to temperature changes compared to Vano et al. (2012). In the analysis of Brikowski (2015), sensitivities of runoff to precipitation and temperature are derived from respective historical data. Thus, sensitivity to temperature will also include the changes caused by the interaction of temperature with other meteorological

variables. In a study with a different approach, Yang and Yang (2011) separated the effect of precipitation, temperature, net radiation, relative humidity and wind speed on runoff and calculated sensitivities for each variable. They reported values of S to temperature ranging from -0.11 to -0.02 $C^{-1}$ between 89 catchments of the EAS region. For the same region, we have computed S to temperature as -0.06 $K^{-1}$, which is included into the literature stated range. Moreover, our S values for radiation, humidity and wind speed are also in good

agreement with Yang and Yang (2011). According to Yang and Yang (2011), S to radiation ranges from -1.9 to -0.3, S to humidity from 0.2 to 1.9 and S to wind speed from -0.8 to -0.1. The range refers to values computed for 89 catchments in the EAS region. Our respective values for this region are -1.53 for radiation,





0.82 for humidity and -0.09 for wind speed. This supports the argument that the large deviations of the sensitivity to temperature between our study and the studies of Vano et al. (2012) and Brikowski (2015), result from interactions in the forcing variables included in the referenced studies.

### 4.3 Sensitivity of runoff to radiation

The reported S to radiation values are higher in absolute terms than S to precipitation values for many of the examined regions and also globally (Table 3). However, according to the findings presented in Section 3.5, precipitation and temperature correspond to higher changes in runoff compared to radiation. That is because high S to radiation results from relatively low $\Delta V$ values, rather than from relatively high $\Delta RF$ values (compared e.g. to precipitation). Small $\Delta V$ for radiation is possibly the consequence of combining shortwave

and longwave radiation to calculate the total bias in radiation, as the two radiation components have inverse signs for most regions (Figure 5).

### 4.4 Sensitivity of runoff to specific humidity at high-latitude regions

Although S to humidity for EAS compares well with literature, unexpectedly high values of S to humidity are found for other regions (5.24 for NEU, 9.58 for NEE, 7.58 for NAS). We performed an extra analysis to

investigate this issue and the basic findings are included in the Supplement of this paper. Very high sensitivity of runoff to H is observed for a specific area, the zone between 70N and 40N latitudes. In that zone, a difference of about 10% in H corresponds to an increase of 40% to 60% in runoff. Investigation of the different fluxes related to runoff production in the model revealed two mechanisms that explain this behavior. First, due to higher humidity, the water vapour deficit of the air is reduced and evapotranspiration is

decreased, thus allowing more of the precipitated water available as runoff. This mechanism explains around one third of the magnitude of reported changes in runoff. The second mechanism happens due to super-saturation of the air, especially during the colder months of the year when the dew point is lower, and includes the condensation and deposition of water vapour (direct transition from vapour to ice). Depositioned water accumulates as snowmass. Snowmass is higher for the raw H run (H has positive biases), which results in

increased snowmelt and thus increased runoff.

Since this experiment was performed with a single LSM, it cannot be concluded whether this behavior is common between the LSMs or is an over-sensitivity of the JULES model. However, it highlights the importance of bias correction for specific humidity for specific regions, where runoff would have been highly overestimated using raw specific humidity as forcing.





### 4.5   Study caveats

An issue that must be considered for the interpretation of the results of this study is that they have been based on a single impact model. As the uncertainty stemming from the selection of the impact model is large (Gudmundsson et al. 2012; Hagemann et al. 2013), it is preferable to use multiple models in order to capture

a wide range of possible results. The effect of the meteorological forcing on a hydrological output is heavily model dependent, as different models employ different concepts and/or equations for the representation of key hydrological processes. This concern has been also discussed by other single model studies on meteorological variables' effects on hydrological outputs (Mizukami et al. 2014; Masaki et al. 2015). Nonetheless, the results of single model studies are useful in giving indicative answers on the issues they

examine and set a basis for the methodology needed for respective multi-model applications.

## 5   Summary and conclusions

The present study examined the effect of the biases in GCM output variables on historical runoff simulations, using the JULES LSM. The effects of biases were studied for each forcing variable separately, for a total of six meteorological variables (precipitation, temperature, radiation, specific humidity, surface pressure and

wind speed). Biases of each variable and the respective effect of runoff were quantified at the global and regional scale. A framework for the categorization of the effects of biases of the different variables was developed and implemented, leading to global maps of bias ECs.

We found that bias correction of GCM outputs results to substantially improved representation of historical runoff. For this reason, our study adds to the numerous studies that advocate on the use of some kind of bias

correction of GCM data prior to their use as impact model forcing. Precipitation and temperature biases were identified to cause the largest changes in runoff. Radiation and specific humidity can also pose a substantial effect on runoff, especially for specific regions. The sensitivity of runoff to the different forcing variables exhibits a high spatial variability. Depending on the region, runoff can be more sensitive to radiation or humidity compared to precipitation or temperature. The produced EC maps show that all variables can

potentially affect runoff to a high extent depending on the region. The fraction of the land surface occupied by the high effect category ECI (high changes in runoff and high sensitivity of runoff to the variable's changes) ranges between the variables from 67.80% for precipitation to 6.09% for wind.

The produced maps of ECs aid the identification of the regions mostly affected by the bias of each variable. Thus, they could serve as a decision tool in cases when an informed decision needs to be made on the

variables that would need to be bias corrected or could be neglected from bias correction, according the planned model application. Moreover, when raw forcing is used in model applications, EC maps could provide a guidance towards the areas where the results would need a more careful interpretation.





Based on the findings of this study we suggest that the widely used concept of bias correcting precipitation and temperature should be extended to include more input variables. Radiation and specific humidity should be added to the priority variables for bias correction in hydrological applications, along with precipitation and temperature.

Due to the heavily model dependent nature of runoff sensitivity to forcing variables, generalized conclusions for the behavior of other impact models to GCM biases cannot be drawn from the present single model assessment. Nevertheless, this study aims to initiate a discussion on the effect of GCM biases on hydrological output, as the consideration of these sensitivities is crucial to understand the uncertainty spectrum of hydrologically relevant climate change assessments.

**Acknowledgements**

We acknowledge the World Climate Research Programme's Working Group on Coupled Modelling, which is responsible for CMIP, and we thank the climate modeling groups (listed in Table 1 of this paper) for producing and making available their model output. For CMIP the U.S. Department of Energy's Program for Climate Model Diagnosis and Intercomparison provides coordinating support and led development of

software infrastructure in partnership with the Global Organization for Earth System Science Portals.

The research leading to these results has received funding from HELIX project of the European Union's Seventh Framework Programme for research, technological development and demonstration under grant agreement no 603864.

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





Table 1. Information on the GCMs used for this study.

| Modelling group | Institute ID | Model name | ºlon x ºlat | Key reference |
|---|---|---|---|---|
| Institut Pierre-Simon Laplace | IPSL | IPSL-CM5A-LR | 3.75 x 1.88 | Dufresne et al. (2013) |
| Japan Agency for Marine-Earth Science and Technology, Atmosphere and Ocean Research Institute (The University of Tokyo), and National Institute for Environmental Studies | MIROC | MIROC-ESM-CHEM | 2.81 x 2.81 | Watanabe et al. (2011) |
| US Dept. of Commerce/NOAA/Geophysical Fluid Dynamics Laboratory | GFDL-NOAA | GFDL-ESM2M | 2.50 x 2.00 | Dunne et al. (2012) |





Table 2. 24 regions of the globe, selected from Giorgi & Bi (2005).

| Region name | Abbreviation |
| --- | --- |
| North Europe | NEU |
| Mediterranean Basin | MED |
| Northeast Europe | NEE |
| North Asia | NAS |
| Central Asia | CAS |
| Tibet | TIB |
| Eastern Asia | EAS |
| Southeast Asia | SEA |
| Northern Australia | NAU |
| Southern Australia | SAU |
| Sahara | SAH |
| Western Africa | WAF |
| Eastern Africa | EAF |
| East Equatorial Africa | EQF |
| South Equatorial Africa | SQF |
| Southern Africa | SAF |
| Western North America | WNA |
| Central North America | CNA |
| Eastern North America | ENA |
| Central America | CAM |
| Amazon | AMZ |
| Central South America | CSA |
| Southern South America | SSA |
| South Asia | SAS |





**Table 3. Relative change (%) in forcing variable (ΔV), corresponding relative change (%) in runoff (ΔRF) and sensitivities (S= ΔRF/ΔV) per region, for each variable. For each region, the median of the ΔV, ΔRF and S values of all land grid boxes is shown.**

|  | Variables | P | T* | R | H | Ps | W |
|---|---|---|---|---|---|---|---|
| **GLOBAL** | ΔV | 14.46 | -0.57 | 1.73 | 0.91 | -0.02 | -5.86 |
|  | ΔRF | 2.49 | 3.38 | -3.71 | 2.04 | -0.04 | 0.21 |
|  | S | 1.76 | -0.05 | -2.12 | 0.81 | 1.18 | -0.06 |
| **NEU** | ΔV | 14.6 | -0.46 | 1.86 | 4.1 | -0.05 | -9.79 |
|  | ΔRF | 27.97 | 22.68 | -5.25 | 25.49 | -0.02 | 3.62 |
|  | S | 2.10 | -0.31 | -3.31 | 5.24 | 2.90 | -0.36 |
| **MED** | ΔV | -14.39 | -0.15 | 0.55 | -1.34 | 0.41 | 14.94 |
|  | ΔRF | -58.56 | 1.55 | -1.51 | 4.07 | 0.44 | -0.47 |
|  | S | 2.02 | -0.04 | -2.52 | 0.77 | 1.08 | -0.08 |
| **NEE** | ΔV | 4.89 | -1.44 | 2.44 | 3.32 | 0.1 | -11.77 |
|  | ΔRF | 5.75 | 47.11 | -5.39 | 32.73 | 0.26 | 5.98 |
|  | S | 2.28 | -0.32 | -2.64 | 9.58 | 3.31 | -0.50 |
| **NAS** | ΔV | 26.05 | 0.67 | 3.53 | 8.05 | -0.06 | -1.08 |
|  | ΔRF | 59.36 | 11.8 | -10.08 | 63.98 | 0.02 | 4.06 |
|  | S | 2.35 | -0.07 | -2.95 | 7.58 | 2.43 | -0.29 |
| **CAS** | ΔV | 6.44 | -0.03 | 1.37 | -13.00 | -0.41 | 8.09 |
|  | ΔRF | -9.94 | 1.31 | -0.44 | -0.19 | -0.36 | -1.29 |
|  | S | 2.49 | -0.05 | -3.50 | 0.31 | 0.88 | -0.09 |
| **TIB** | ΔV | 128.47 | -2.94 | -1.14 | 7.69 | -0.12 | 12.59 |
|  | ΔRF | 1017.17 | 5.38 | 0.97 | 0.81 | 0.02 | 0.06 |
|  | S | 7.27 | -0.02 | -2.07 | 0.18 | 0.40 | 0.00 |
| **EAS** | ΔV | 19.25 | -0.94 | 2.51 | 2.92 | -0.2 | -3.55 |
|  | ΔRF | 4.36 | 5.54 | -2.96 | 3.66 | -0.05 | 0.76 |
|  | S | 1.70 | -0.06 | -1.53 | 0.82 | 1.07 | -0.09 |
| **SEA** | ΔV | 19.76 | -0.87 | 1.11 | 0.89 | 0.23 | 34.57 |
|  | ΔRF | 43.92 | 5.97 | -3.2 | 1.66 | 0.32 | -1.04 |
|  | S | 2.07 | -0.08 | -2.68 | 1.16 | 1.54 | -0.05 |
| **NAU** | ΔV | 41.15 | -0.04 | 1.43 | 7.71 | 0.1 | -28.46 |





|  |  |  |  |  |  |  |  |
|---|---|---|---|---|---|---|---|
|  | ΔRF | -5.13 | 1.02 | -1.16 | 1.38 | 0.09 | -0.44 |
|  | S | 0.37 | -0.03 | -0.75 | 0.31 | 0.56 | 0.00 |
| **SAU** | ΔV | 18.92 | -0.28 | 0.85 | 2 | -0.13 | -11.2 |
|  | ΔRF | -9.29 | 1.07 | -0.11 | 1.4 | 0.06 | -0.49 |
|  | S | 0.82 | -0.05 | -0.88 | 0.67 | 1.00 | -0.03 |
| **SAH** | ΔV | 54.11 | -2.73 | -0.47 | -8.96 | 0.22 | -13.59 |
|  | ΔRF | -2.59 | -0.68 | 0.64 | -0.32 | 0 | 0.08 |
|  | S | 0.94 | 0.00 | -0.25 | 0.04 | 0.04 | -0.01 |
| **WAF** | ΔV | 26.74 | -1.51 | -0.88 | -5.79 | -0.1 | -15.13 |
|  | ΔRF | 58.24 | 5.61 | -1.57 | -0.71 | -0.13 | 0.09 |
|  | S | 2.78 | -0.04 | -2.61 | 0.22 | 1.28 | -0.04 |
| **EAF** | ΔV | 23.22 | -1.68 | -0.06 | -5.76 | -0.25 | -12.11 |
|  | ΔRF | 42.13 | 7.24 | -1.51 | -3.74 | -0.28 | 0.09 |
|  | S | 2.12 | -0.05 | -1.95 | 0.48 | 0.95 | 0.00 |
| **EQF** | ΔV | 5.64 | -1.55 | -0.25 | -2.15 | -0.2 | -10.09 |
|  | ΔRF | -0.14 | 6.21 | 0.92 | -1.29 | 0 | 0.07 |
|  | S | 2.26 | -0.05 | -1.73 | 0.49 | 0.92 | -0.01 |
| **SQF** | ΔV | 36.45 | -0.9 | 0.9 | 0.89 | -0.03 | -15.6 |
|  | ΔRF | -73.18 | -82.26 | -85.07 | -84.68 | -84.2 | -84.18 |
|  | S | 2.94 | -0.07 | -1.91 | 0.59 | 1.10 | -0.04 |
| **SAF** | ΔV | 89.8 | -1.41 | -0.38 | 14.28 | 0.68 | -4.74 |
|  | ΔRF | 85.47 | 5.5 | 0.54 | 5.33 | 0.42 | -0.02 |
|  | S | 1.35 | -0.04 | -1.66 | 0.45 | 0.72 | -0.05 |
| **WNA** | ΔV | 65.92 | -1.75 | -1.23 | 13.55 | 0.14 | 10.23 |
|  | ΔRF | 112.66 | 17.94 | -0.48 | 9.85 | 0.16 | -2.5 |
|  | S | 2.12 | -0.13 | -2.01 | 0.77 | 0.98 | -0.17 |
| **CNA** | ΔV | -12.84 | 0.11 | 1.68 | 2.29 | -0.08 | -14.79 |
|  | ΔRF | -50.86 | 1.53 | -2.06 | 6.57 | -0.05 | 1.96 |
|  | S | 2.54 | -0.07 | -1.47 | 1.08 | 1.09 | -0.13 |
| **ENA** | ΔV | 4.08 | 0.49 | 2.71 | 13.4 | 0.1 | 5.47 |




|      |     |        |       |       |        |       |       |
|------|-----|--------|-------|-------|--------|-------|-------|
|      | ΔRF | -0.38  | -0.38 | -5.18 | 39.72  | 0.13  | 0.86  |
|      | S   | 1.69   | -0.07 | -1.92 | 3.17   | 1.54  | -0.11 |
| **CAM** | ΔV  | 11.43  | -0.98 | -0.4  | -6.16  | 0.15  | 25.27 |
|      | ΔRF | -7.73  | 3.65  | -0.1  | -2.55  | 0.14  | -0.52 |
|      | S   | 1.32   | -0.04 | -1.58 | 0.49   | 0.77  | -0.02 |
| **AMZ** | ΔV  | -26.58 | -0.35 | 4.06  | -13.19 | -0.19 | -4    |
|      | ΔRF | -40.52 | 4.88  | -9.34 | -6.01  | -0.23 | 0.03  |
|      | S   | 1.42   | -0.05 | -2.37 | 0.53   | 1.44  | -0.04 |
| **CSA** | ΔV  | -32.8  | 0.7   | 3.05  | -11.53 | -0.23 | -7.5  |
|      | ΔRF | -63.21 | -1.49 | -3.22 | -5.75  | -0.13 | 0.38  |
|      | S   | 1.59   | -0.04 | -1.16 | 0.53   | 0.83  | -0.04 |
| **SSA** | ΔV  | 72.07  | -1.22 | -1.77 | 5.07   | 0.08  | 9.91  |
|      | ΔRF | 84.32  | 10.06 | -0.47 | 12.05  | 0.34  | -2.44 |
|      | S   | 1.53   | -0.09 | -0.50 | 1.48   | 1.29  | -0.04 |
| **SAS** | ΔV  | -9.19  | -1.08 | 1.39  | -13.11 | -0.05 | -6.81 |
|      | ΔRF | -26.35 | 5.2   | -4.07 | -2.53  | -0.09 | 0.51  |
|      | S   | 1.62   | -0.05 | -2.46 | 0.29   | 0.90  | -0.05 |

*ΔV for temperature is the absolute change in temperature.



**Table 4. Percent of land area (%) under each of the four Effect Categories (ECs).**

| Variables / ECs | I | II | III | IV |
|---|---|---|---|---|
| **P** | 67.80 | 24.20 | 1.82 | 6.18 |
| **T** | 45.15 | 22.03 | 2.46 | 30.35 |
| **R** | 48.74 | 1.30 | 26.16 | 23.80 |
| **H** | 40.80 | 13.76 | 5.58 | 39.86 |
| **Ps** | 12.17 | 1.83 | 38.48 | 47.52 |
| **W** | 6.09 | 19.19 | 2.35 | 72.37 |




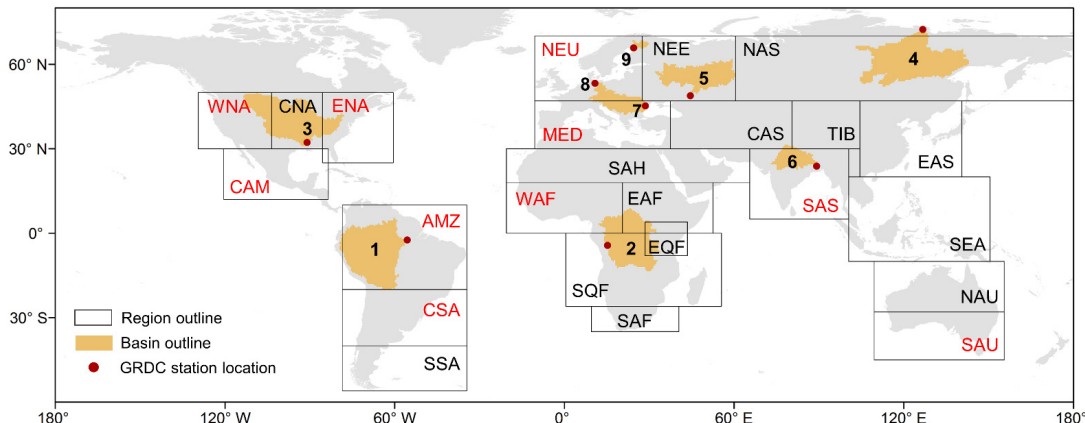

**Figure 1. Outlines of study focus regions and hydrological basins and locations of the GRDC gauging stations. With red colour are noted the regions selected for more detailed analysis. The hydrological basins have been numbered in decreasing order according to their area: 1) Amazon, 2) Congo, 3) Mississippi, 4) Lena, 5) Volga, 6) Ganges, 7) Danube, 8) Elbe, 9) Kemijoki.**





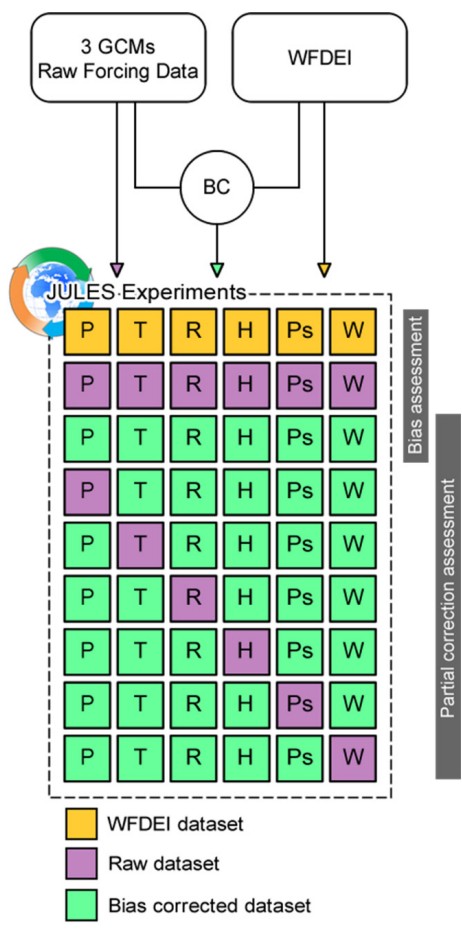

**Figure 2. Graphical description of the performed experiment.**



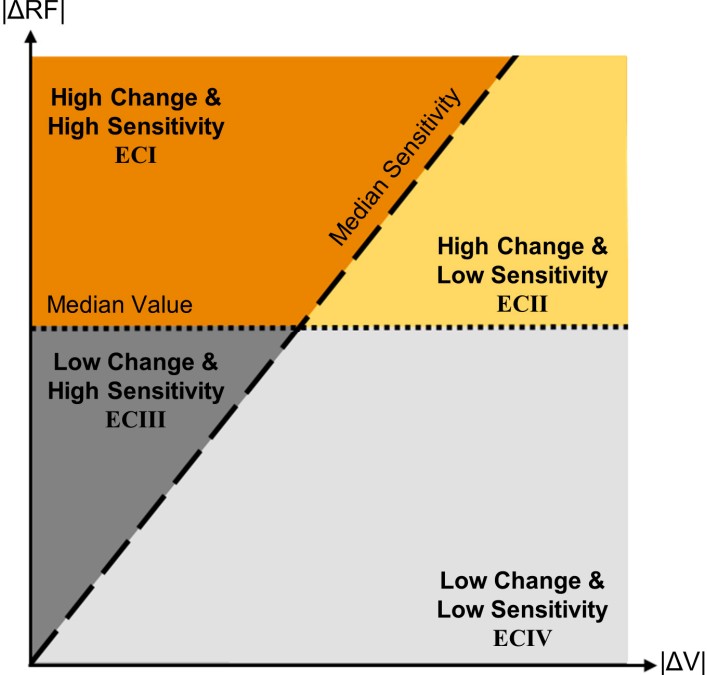

**Figure 3. Categorization of the effect of changes in forcing variables (V) on runoff (RF). The four areas correspond to the four defined Effect Categories. The x axis corresponds to relative changes in forcing variables and the y axis to relative changes in runoff. For all changes, the absolute value is considered.**





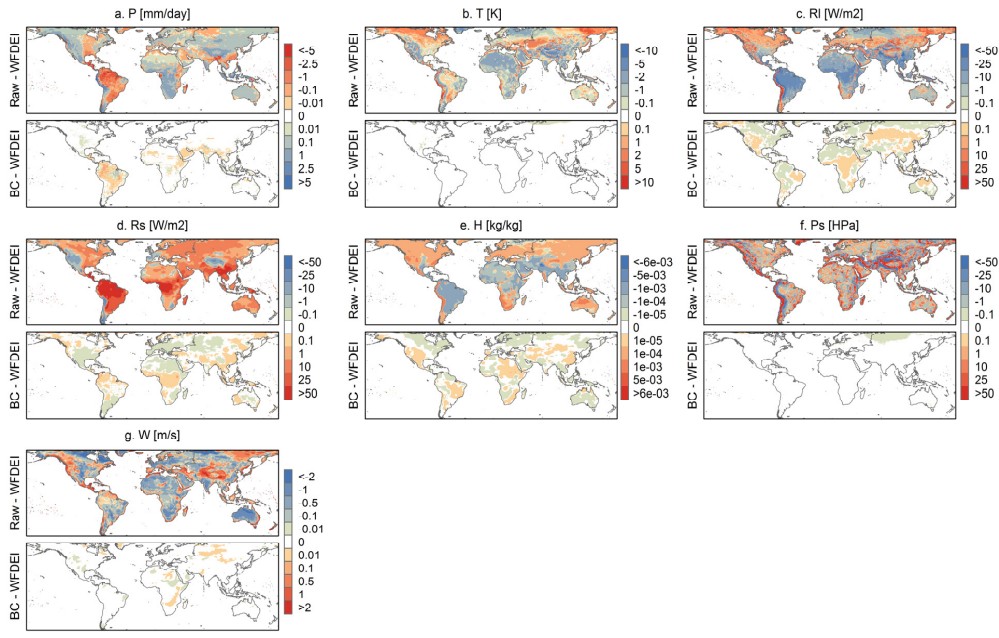

**Figure 4. Difference maps, showing initial (Raw-WFDEI) and remaining (BC-WFDEI) biases of the GCM ensemble forcing variables: a. Precipitation, b. Temperature, c. Longwave downward radiation, d. Shortwave downward radiation, e. Specific humidity, h. Surface pressure, g. Wind. Differences are calculated between the long-term annual averages (ANN) of the 1981-2010 period.**





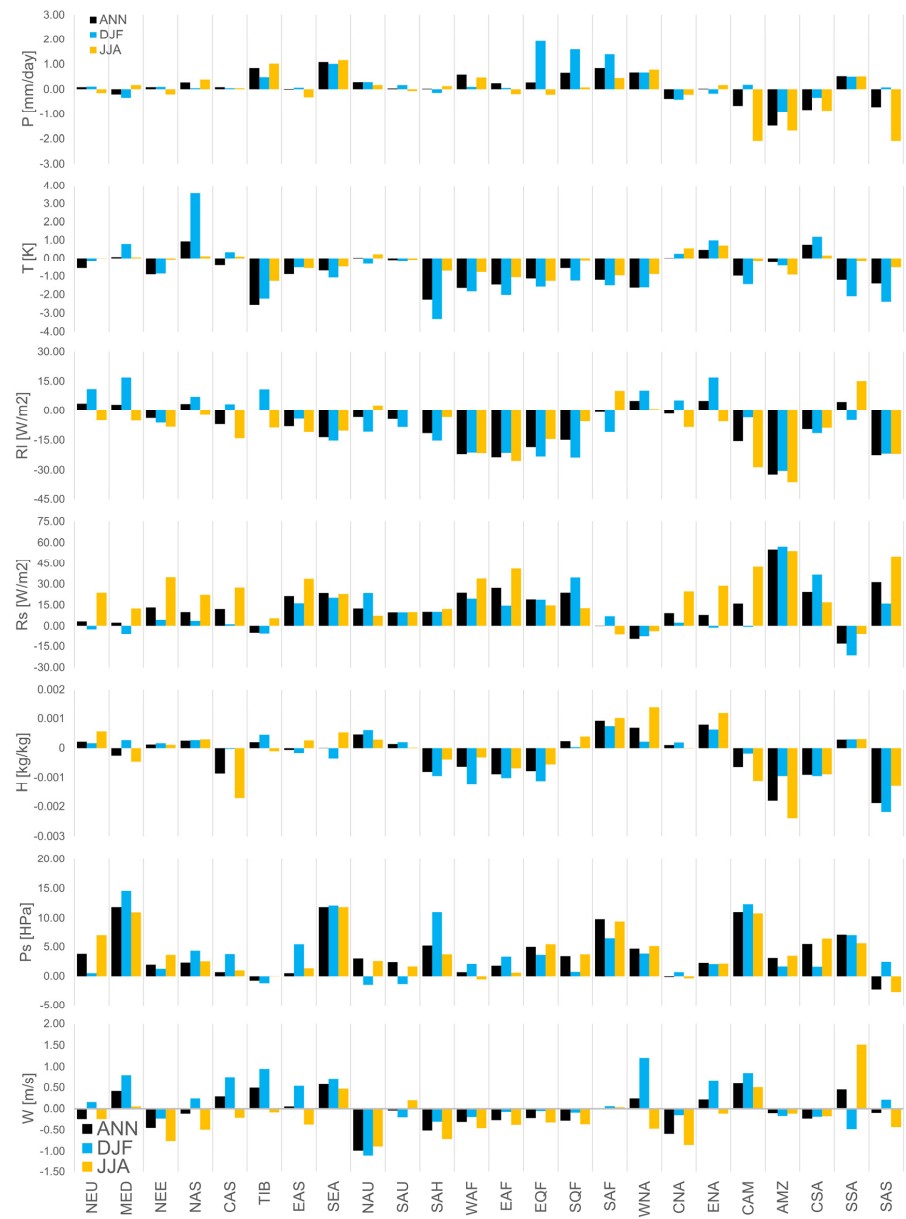

**Figure 5. Initial biases (Raw-WFDEI) of the GCM ensemble forcing variables, spatially averaged for 24 Giorgi regions. Biases are calculated between long-term annual averages (ANN), December-January-February (DJF) and June-July-August (JJA) averages of the period 1981-2010.**





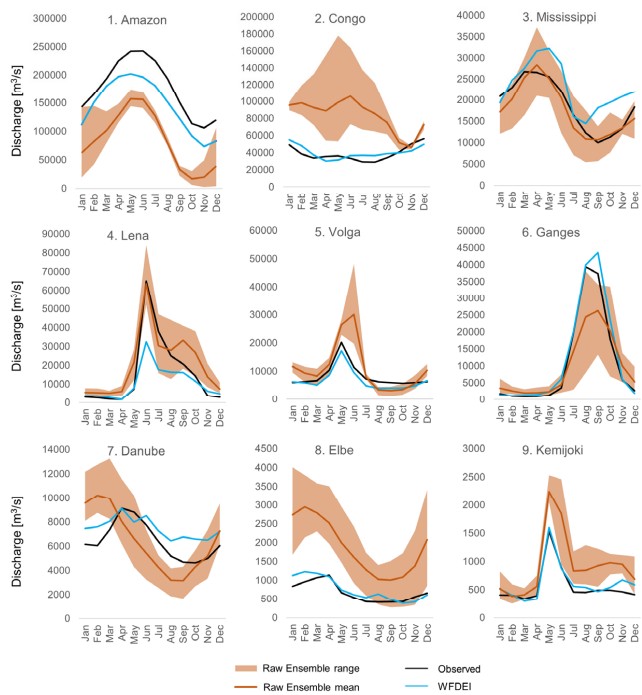

**Figure 6. Discharge seasonality [m³/s] derived from the period 1981-2010 for 9 study basins. Each panel shows observed discharge (GRDC measurements) compared to JULES' simulated discharge from WFDEI data and Raw GCM data (the mean and the range of the ensemble are shown).**





| Indices | NSE | | PBIAS [%] | | R² | |
|---|---|---|---|---|---|---|
| Basins | WFDEI | Raw EM | WFDEI | Raw EM | WFDEI | Raw EM |
| Amazon | 0.48 | -2.66 | -18.68 | -51.84 | 0.96 | 0.94 |
| Congo | 0.39 | -36.40 | 4.06 | 116.77 | 0.45 | 0.20 |
| Mississippi | 0.24 | 0.90 | 21.56 | -4.46 | 0.73 | 0.92 |
| Lena | 0.56 | 0.82 | -39.32 | 32.14 | 0.98 | 0.89 |
| Volga | 0.82 | -1.42 | -17.09 | 35.12 | 0.95 | 0.66 |
| Ganges | 0.94 | 0.80 | 19.48 | -9.51 | 0.99 | 0.91 |
| Danube | 0.28 | -1.51 | 15.20 | 1.14 | 0.88 | 0.19 |
| Elbe | 0.67 | -26.04 | 8.28 | 179.83 | 0.81 | 0.86 |
| Kemijoki | 0.91 | -0.98 | 8.55 | 66.50 | 0.94 | 0.89 |

Worst fit         NSE=0         Best fit
|PBIAS|=25%
R²=0.5

**Figure 7. Comparison of evaluation metrics derived from monthly discharge data. Metrics are calculated for JULES' simulations from WFDEI data (WFDEI) and the ensemble mean of Raw GCM data (Raw EM). The color hue corresponds to "good" or "bad" model fits to observations. The assessment is done for each metric separately. The following rules have been set to describe "good"**
5   **performance (per metric): NSE>0, |PBIAS|<25% and R²>0.5.**



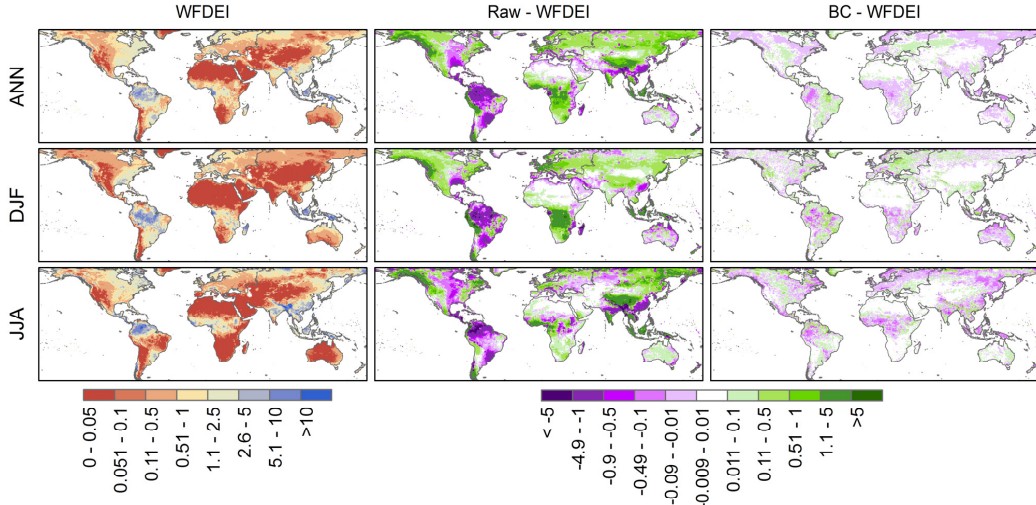

**Figure 8. Runoff [mm/day], from WFDEI data (left column). Initial (Raw-WFDEI) and remaining (BC-WFDEI) biases in runoff are shown in middle and right columns respectively. Results are shown for long-term annual averages (ANN), December-January-February (DJF) and June-July-August (JJA) averages of the 1981-2010 period.**



**Figure 9. (top row)** Runoff [mm/day], from bias corrected GCM ensemble forcing (BC), and (second to last row) runoff differences between the bias corrected run (BC) and the partially corrected runs (NobcV, where V is one of the forcing variables P, T, R, H, Ps, W). Results are shown for long-term annual averages (ANN), December-January-February (DJF) and June-July-August (JJA) averages of the 1981-2010 period.





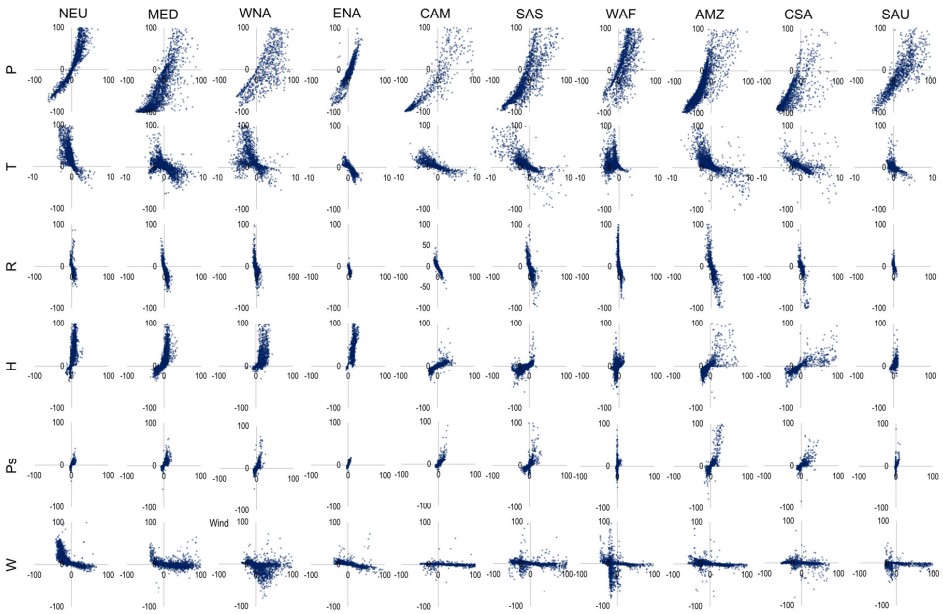

**Figure 10. Scatterplots of relative changes in forcing variable (ΔV, x axis) and corresponding relative changes in runoff (ΔRF, y axis), for all the forcing variables and for selected regions. In each panel, each dot represents the ΔRF/ ΔV relationship of each land grid box in the examined region.**



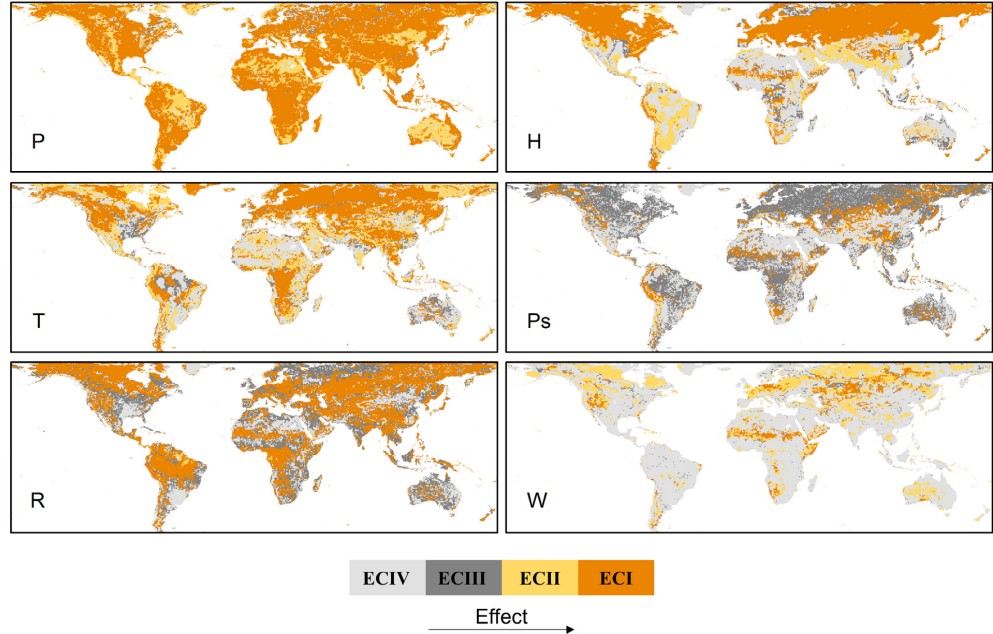

**Figure 11. Global maps of bias Effect Categories (ECs) for each forcing variable.**