# Peer review of "The effect of GCM biases on global runoff simulations of a land surface model"

_Hydrology and Earth System Sciences, 2017_

## Referee Comment (RC1) · Anonymous Referee #1 · 7 May 2017

Papadimitriou et al. submitted the manuscript "The effect of GCM biases on global runoff simulations of a land surface model" to Hydrology and Earth System Sciences as a revised version of the manuscript "Hotspots of sensitivity to GCM biases in global modeling of mean and extreme runoff" (doi:10.5194/hess-2016-547 (hereafter called as HESS16)). Main focus of the manuscript is the assessment of GCM biases to the impact model JULES and runoff. Compared with the HESS16 version, this manuscript is presented much clearer and consistent and many of the referee suggestions were considered. I therefore acknowledge the authors for carefully revising the manuscript. However, there are a few issues that needs to be addressed before publication.

Major

In all the analyses, the ensemble mean (of 3 GCMs) is shown, but it would be very

informative to see exemplarily the behavior of the single GCMs within the focus of the study. This also affects the question about the reason to select the specific 3 GCMs out of CMIP5. For example, I am surprised to see the huge difference of "Raw – WFDEI" for Rs and Rl in Fig. When I am interpreting the Rs color values correct, the ensemble GCM are > 50 W m-2 higher for nearly complete South America (and the other way around for Rl). Is that consistent among the GCMs? I realize the range of the raw GCM range in Fig. 6, esp. for Congo. In order to see the effect of bias correction, please consider drawing also results for the bias corrected GCM runs. Sure, this adds another color, but this figure can also be redrawn showing e.g. with 3 red lines for each raw GCM and 3 green lines for each bias corrected GCM. This would provide the reader a much more visual interpretation of the effect of bias correction to discharge seasonality and could be an added value of the overall study.

Structural, the paper misses a clear separation between "Results" and "Discussion". For example, section "The model evaluation..." at page 11 reads for me like a discussion (finding out reasons for performance of the model). Please move to discussion part. Another example is Page 13 section starting with "First.." – the authors itself write that they discuss. Please avoid that in a results section. Similar difficulties I have with P15, section starting with "The variation..". You could also consider to have a joint "Results and discussion".

Minor

P5, l16: "been used in the BCIP": Could you please write some essentials of the inter-comparison results? This is especially of importance, as it seems that the method is only applied in studies by the authors of this manuscript.

P5, l28: could you somehow describe "edge segments" e.g. by a percentile? Otherwise it reads a bit vague.

P8, l7: what is meant by median value? Of each grid cell in a specific region? Please specify.

P11 section starting with "The shown persistent...": Although interesting, it is a bit vague and could be supported by citing common papers (e.g. Coxon et al. (2015) for discharge uncertainty). Please either move the last sentence "We believe..." to discussion and discuss it properly or just delete it. It is too speculating without providing reasons for this statement.

P12, l12: please check the statement, that global LSMs are calibrated (the sentence reads so).

P16, l15: I cannot see terrain elevation in Fig. 11, so the statement cannot be made (strictly speaking).

Please go carefully through the reference list. A quick look on it shows a lot of inconsistencies. For example: first reference – Journal name is missing – and why are you citing the discussion paper and not the final version? Check consistency of Giorgi and Bi. Check if everywhere a doi is provided, check if upper/lower case is consistent, Hattermann et al 2016 is published since a while (please update citation), and what does "Submitted in this special issue" should mean? N/a in Maraun 2012? What are "and Ohters" in Nikulin et al?, Journal / doi for Oki and Sud? I did not check if all references are listed in the reference list / in the manuscript.

Figure 7: I am a bit sceptic to consider a NSE > 0 as "good". In many studies, this is the case of e.g. > 0.5 or 0.7. It is absolutely not necessary to provide color hues to indicate how good or bad a model performs, it hinders (me) for an objective look at the table (in fact, it is a table with colored cells). Please convert to a real table for more clarity.

Figure 11: Please consider other colors to distinguish ECII and ECI for better visualization. Supplement:

Table S1 is not referred from the main manuscript, please also provide station name. Fig. S3 – what does the red color mean? Please indicate in figure caption. Figs.

S4-S10 are not mentioned in the main manuscript. To my knowledge, a supplement should support the main paper, and that are interesting figures, but without referencing it in the main paper, they are unconnected and lost with just the figure caption.

Technical

P1, l26: please be consistent: either Global Hydrological Model or global hydrological model, not global Hydrological Model.

P3, l29: check if Penman (1948) is the correct citation for Penman-Monteith approach (isn't it Monteith 1965?)

P5, l 4 (the two sentence starting with "The WFDEI"...): I feel this information is not required for the manuscript. Consider to shorten it.

P6, l31: please be as specific as possible in naming the variables. Is it net shortwave radiation, or downward shortwave / longwave (which I am sure is meant?)

P12, l26: please insert a blank between number and unit. Same at l30 (5mm)

P16, l6: to which section are you referring to?

References

Coxon, G., Freer, J., Westerberg, I. K., Wagener, T., Woods, R. and Smith, P. J.: A novel framework for discharge uncertainty quantification applied to 500 UK gauging stations, Water Resour. Res., 51(7), 5531–5546, doi:10.1002/2014WR016532, 2015.

---

## Referee Comment (RC2) · Anonymous Referee #2 · 25 May 2017

General comments:

This paper examines the impact of bias correction on global uncoupled land-surface model (LSM) simulations using GCM data as inputs. They step through an observation-based simulation, and two GCM simulations using raw GCM inputs and a bias correction simulation that corrects all seven variables required to run an LSM that solves the surface energy balance. They find the bias correction scheme removes nearly all bias from all seven variables and the subsequent LSM simulations closely match the observation-based simulation.

They then step through six simulations where they remove bias correction from one variable at a time (e.g. precipitation or temperature) to examine the impact of bias correction for that specific variable. Findings highlight that precipitation and temperature

are the most important variables to bias correct, as expected. However, radiation is also found to have high sensitivity and should be corrected as well. Surface pressure and wind speed generally have little impact and can almost always be neglected. I appreciate the literature review to place the sensitivities found here in the context of other studies.

Extra time is spent examining humidity, as extreme sensitivity across higher latitudes in the northern hemisphere is found in the LSM (JULES in this paper). They attribute the extreme sensitivity to reduced ET in high humidity environments, and direct condensation and deposition of water vapor into the snowpack in these regions.

Overall, this paper is logically organized and generally easy to follow. I think it will be ready for publication in HESS after the authors address my comments.

Comments:

1) The specific humidity discussion is much needed to explain the extreme sensitivities in the JULES model. I believe it should be expanded in the main text, with the figures in the supplemental material moved to the paper. Additionally, examination of relative humidity for supersaturated conditions should be done for an example grid cell or the entire Northern European region.

Figure S.4 and the discussion in Section 4.4 imply that the raw GCM runs are too humid and have supersaturated conditions in high latitudes in the northern Hemisphere. I'm a bit surprised by this, thus it is worth checking in more detail by looking at relative humidity over a larger spatial region in the raw GCM output.

What GCM data are used to force JULES? That is a key detail I cannot find in the paper. Was it the lowest sigma level of the GCM, or the 10-m/2-m diagnosed variables from the GCM? It is possible (although maybe not likely) that the issue could stem from the use of JULES while the GCMs use a different LSM and that mismatch could be at the root of these results.

Finally, some detailed examination of how JULES treats vegetation and snowpack versus the LSMs in the GCMs may also give some insight into this issue. These regions are primarily boreal forest, which are sometimes difficult areas to model in winter, particularly surface fluxes into/out of the snowpack (e.g. Chen et al. 2014). Is it possible to distinguish between condensation and deposition in the JULES output? That may give additional insight into JULES behavior.

2) Discussion in sections 3.1 and 3.2 can be shortened to make room for the expanded discussion of the specific humidity biases. These add little value to the paper in my opinion.

3) The bias correction procedure needs more detail discussed. The authors mention the issues of keeping variables physically consistent when performing bias correction across many variables. However, they do not discuss the basic checks that are still necessary and should be performed when bias correcting many variables. For example, limiting humidity to prevent supersaturation conditions in the bias corrected fields when correcting temperature and specific humidity.

References: Chen, F., et al. (2014), Modeling seasonal snowpack evolution in the complex terrain and forested Colorado Headwaters region: A model intercomparison study, J. Geophys. Res. Atmos., 119, 13,795–13,819, doi:10.1002/2014JD022167.

---

## Author Comment (AC1) · 20 Jun 2017

Please refer to the attached supplement

Please also note the supplement to this comment:
http://www.hydrol-earth-syst-sci-discuss.net/hess-2017-208/hess-2017-208-AC1-supplement.pdf

---

## Author Response (AR1)

We would like to thank Anonymous Referee #1 for providing us with constructive comments on the submitted manuscript. We believe that the input that will help improve the manuscript significantly. We would also like to thank the reviewer for acknowledging the improvement of the present manuscript compared to the previously submitted HESS16. The reviewer's comments (shown in italics) have been addressed point-by-point.

Anonymous Referee #1

*Papadimitriou et al. submitted the manuscript "The effect of GCM biases on global runoff simulations of a land surface model" to Hydrology and Earth System Sciences as a revised version of the manuscript "Hotspots of sensitivity to GCM biases in global modeling of mean and extreme runoff" (doi:10.5194/hess-2016-547 (hereafter called as HESS16)). Main focus of the manuscript is the assessment of GCM biases to the impact model JULES and runoff. Compared with the HESS16 version, this manuscript is presented much clearer and consistent and many of the referee suggestions were considered. I therefore acknowledge the authors for carefully revising the manuscript. However, there are a few issues that needs to be addressed before publication.*

*Major*

*RC1. In all the analyses, the ensemble mean (of 3 GCMs) is shown, but it would be very informative to see exemplarily the behavior of the single GCMs within the focus of the study. This also affects the question about the reason to select the specific 3 GCMs out of CMIP5. For example, I am surprised to see the huge difference of "Raw – WFDEI" for Rs and Rl in Fig 4. When I am interpreting the Rs color values correct, the ensemble GCM are > 50 W m-2 higher for nearly complete South America (and the other way around for Rl). Is that consistent among the GCMs?*

AC1. To give an insight into the behavior of each ensemble member, we estimated the spatial averages of the raw input variables for each single GCM over the study regions. This information, along with the respective ensemble mean and WFDEI values is summarized in a Table added in the Supplement of this paper (Table S2) and also in the following pages of this reply. The authors believe that this information can help clarify the source of possibly large biases presented in Figure 4.

Specifically on the example given by the reviewer, concerning the initial biases of Rs and Rl in Figure 4, an examination of values for the AMZ region can provide some relevant answers. Comparison of the single GCM values with WFDEI (for AMZ) indicate that: for Rl the initial biases are relatively consistent among the 3 GCMs, while for Rs, IPSL has the largest contribution to the ensemble mean initial bias , as its difference from WFDEI is double compared to the other two GCMs. A note on the color interpretation of Figure 4 mentioned by the reviewer: "Raw-WFDEI" for Rs is in the greatest value class (>50 W/m$^2$) for the northern part of South America and in the second greatest class (25 to 50 W/m$^2$) for central South

America. For Rl, most of South America falls into the second greatest underestimation class (-25 to -50 W/m$^2$).

The selected GCMs have participated in the ISIMIP Fast Track experiment. The 3 GCMs cover the full range of the climate sensitivity along the 5 Fast Track experiment models, hence it was decided to limit the ensemble to 3 members due to the large number of runs needed for the experiment conducted in this study.

*RC2. I realize the range of the raw GCM range in Fig. 6, esp. for Congo. In order to see the effect of bias correction, please consider drawing also results for the bias corrected GCM runs. Sure, this adds another color, but this figure can also be redrawn showing e.g. with 3 red lines for each raw GCM and 3 green lines for each bias corrected GCM. This would provide the reader a much more visual interpretation of the effect of bias correction to discharge seasonality and could be an added value of the overall study.*

AC2. The bias corrected results are not presented in Figure 6 for the sake of visual clarity, as bias corrected results are very close to each other and to the WFDEI results, and their values are almost indistinguishable. Following the reviewer's indication, a new figure has been produced to include the bias corrected results and is presented below. However, the authors feel that the new figure does not provide substantial new information due to the almost indistinguishable behavior of the 3 bias corrected GCMs -and it is also more difficult to interpret- and suggest that the previous version of the figure should remain in the manuscript.

[Figure]

**Figure 6. Discharge seasonality [m³/s] derived from the period 1981-2010 for 9 study basins. Each panel shows observed discharge (GRDV measurements) compared to JULES' simulated discharge from WFDEI data, raw GCM data and bias corrected GCM data.**

**Table S 1. Values of input variables, for each GCM (GFDL, IPSL and MIROC), the ensemble mean (Ens.Mean) and WFDEI data, spatially averaged for 24 Giorgi regions.**

| | P [mm/day] | | | | | T [K] | | | | |
|---|---|---|---|---|---|---|---|---|---|---|
| | *GFDL* | *IPSL* | *MIROC* | *Ens.Mean* | *WFDEI* | *GFDL* | *IPSL* | *MIROC* | *Ens.Mean* | *WFDEI* |
| **NEU** | 2.61 | 2.30 | 2.53 | 2.48 | 2.43 | 277.90 | 277.15 | 281.16 | 278.74 | 279.50 |
| **MED** | 1.44 | 1.08 | 1.44 | 1.32 | 1.56 | 288.73 | 287.57 | 290.13 | 288.81 | 288.26 |
| **NEE** | 1.71 | 1.67 | 1.79 | 1.72 | 1.67 | 274.42 | 274.15 | 277.89 | 275.49 | 276.75 |
| **NAS** | 1.59 | 1.66 | 1.78 | 1.68 | 1.25 | 267.91 | 269.70 | 270.71 | 269.44 | 267.53 |
| **CAS** | 0.92 | 0.79 | 1.36 | 1.02 | 0.93 | 284.84 | 284.00 | 287.60 | 285.48 | 285.79 |
| **TIB** | 1.36 | 1.05 | 1.99 | 1.47 | 0.63 | 274.44 | 271.79 | 273.20 | 273.14 | 275.70 |
| **EAS** | 2.96 | 2.88 | 2.96 | 2.94 | 2.57 | 286.26 | 285.73 | 288.39 | 286.79 | 284.48 |
| **SEA** | 8.77 | 6.74 | 6.80 | 7.44 | 6.96 | 299.45 | 299.15 | 298.99 | 299.19 | 299.21 |
| **NAU** | 2.97 | 1.37 | 3.46 | 2.60 | 1.65 | 297.80 | 297.47 | 298.36 | 297.87 | 297.40 |
| **SAU** | 1.79 | 1.60 | 2.28 | 1.89 | 1.28 | 289.28 | 286.59 | 287.49 | 287.79 | 290.68 |
| **SAH** | 0.22 | 0.06 | 0.35 | 0.21 | 0.15 | 297.02 | 294.15 | 296.73 | 295.97 | 298.18 |
| **WAF** | 4.60 | 2.92 | 4.02 | 3.85 | 2.86 | 298.65 | 298.50 | 299.70 | 298.95 | 300.57 |
| **EAF** | 2.15 | 1.52 | 2.87 | 2.18 | 1.99 | 297.86 | 297.14 | 298.09 | 297.69 | 298.99 |
| **EQF** | 2.87 | 3.34 | 2.80 | 3.00 | 2.67 | 295.21 | 295.47 | 295.61 | 295.43 | 296.00 |
| **SQF** | 3.33 | 3.18 | 2.79 | 3.10 | 3.04 | 295.89 | 295.95 | 296.37 | 296.07 | 295.96 |
| **SAF** | 2.37 | 1.62 | 2.20 | 2.06 | 1.27 | 291.60 | 290.33 | 290.83 | 290.92 | 290.89 |
| **WNA** | 1.92 | 1.88 | 2.32 | 2.04 | 1.49 | 282.01 | 282.41 | 284.29 | 282.90 | 282.96 |
| **CNA** | 2.48 | 2.11 | 2.12 | 2.23 | 2.62 | 283.22 | 283.91 | 286.66 | 284.59 | 284.58 |
| **ENA** | 3.53 | 3.49 | 3.77 | 3.60 | 3.20 | 286.57 | 287.57 | 289.45 | 287.86 | 282.26 |
| **CAM** | 3.43 | 2.17 | 2.22 | 2.60 | 2.84 | 295.70 | 295.89 | 297.40 | 296.33 | 295.32 |
| **AMZ** | 3.57 | 3.55 | 4.06 | 3.72 | 5.32 | 297.74 | 297.44 | 297.66 | 297.61 | 297.94 |
| **CSA** | 2.37 | 1.71 | 2.20 | 2.09 | 2.83 | 291.79 | 290.06 | 291.07 | 290.97 | 290.61 |
| **SSA** | 2.58 | 2.76 | 2.70 | 2.68 | 2.57 | 281.71 | 278.10 | 279.75 | 279.85 | 281.32 |
| **SAS** | 3.61 | 2.94 | 4.76 | 3.77 | 3.75 | 296.89 | 296.78 | 297.21 | 296.96 | 296.36 |
| | Rl [W/m2] | | | | | Rs [W/m2] | | | | |
| | *GFDL* | *IPSL* | *MIROC* | *Ens.Mean* | *WFDEI* | *GFDL* | *IPSL* | *MIROC* | *Ens.Mean* | *WFDEI* |
| **NEU** | 298.76 | 289.91 | 313.39 | 300.69 | 295.33 | 106.90 | 113.95 | 105.91 | 108.92 | 115.03 |
| **MED** | 325.96 | 306.36 | 328.73 | 320.35 | 314.19 | 194.08 | 207.62 | 202.13 | 201.27 | 199.11 |
| **NEE** | 283.96 | 268.51 | 293.05 | 281.84 | 286.82 | 113.46 | 130.74 | 131.76 | 125.32 | 113.86 |
| **NAS** | 255.12 | 250.35 | 261.36 | 255.61 | 245.13 | 115.27 | 125.07 | 132.40 | 124.24 | 117.66 |
| **CAS** | 294.43 | 276.17 | 300.68 | 290.43 | 295.95 | 208.62 | 212.39 | 224.00 | 215.01 | 204.59 |
| **TIB** | 254.00 | 226.63 | 239.74 | 240.12 | 239.85 | 193.41 | 203.32 | 238.30 | 211.68 | 216.40 |
| **EAS** | 330.69 | 311.34 | 329.96 | 324.00 | 310.11 | 175.70 | 203.77 | 197.67 | 192.38 | 171.51 |
| **SEA** | 412.92 | 398.55 | 404.30 | 405.26 | 415.89 | 217.69 | 235.62 | 220.55 | 224.62 | 194.56 |
| **NAU** | 375.94 | 353.13 | 375.57 | 368.22 | 357.89 | 245.74 | 275.31 | 245.39 | 255.48 | 248.10 |
| **SAU** | 330.27 | 314.19 | 326.86 | 323.77 | 326.54 | 197.93 | 190.86 | 185.11 | 191.30 | 216.98 |
| **SAH** | 337.31 | 309.98 | 339.92 | 329.07 | 337.15 | 262.15 | 275.38 | 277.74 | 271.75 | 264.56 |

| | | | | | | | | | | |
|---|---|---|---|---|---|---|---|---|---|---|
| **WAF** | 384.32 | 363.56 | 388.70 | 378.86 | 392.92 | 230.64 | 281.46 | 240.12 | 250.74 | 231.51 |
| **EAF** | 371.89 | 347.30 | 372.53 | 363.91 | 384.45 | 251.09 | 292.60 | 247.54 | 263.74 | 237.33 |
| **EQF** | 372.31 | 356.07 | 365.27 | 364.55 | 377.08 | 240.21 | 278.16 | 231.80 | 250.05 | 232.56 |
| **SQF** | 378.02 | 362.43 | 370.00 | 370.15 | 373.27 | 234.04 | 268.65 | 237.10 | 246.60 | 223.85 |
| **SAF** | 344.64 | 323.67 | 334.37 | 334.23 | 321.71 | 217.70 | 237.28 | 219.01 | 224.66 | 232.14 |
| **WNA** | 296.89 | 293.37 | 302.39 | 297.55 | 281.30 | 196.70 | 183.22 | 195.71 | 191.87 | 205.10 |
| **CNA** | 311.69 | 298.60 | 310.79 | 307.03 | 308.70 | 178.09 | 198.56 | 207.13 | 194.59 | 185.28 |
| **ENA** | 339.03 | 327.43 | 341.57 | 336.01 | 305.46 | 171.46 | 189.71 | 187.69 | 182.95 | 164.46 |
| **CAM** | 377.27 | 360.63 | 370.16 | 369.35 | 366.67 | 229.89 | 252.57 | 248.63 | 243.70 | 229.00 |
| **AMZ** | 386.81 | 370.84 | 385.43 | 381.03 | 410.20 | 236.57 | 276.72 | 229.83 | 247.71 | 195.18 |
| **CSA** | 345.94 | 327.65 | 331.53 | 335.04 | 336.63 | 213.80 | 221.64 | 223.21 | 219.55 | 210.34 |
| **SSA** | 306.49 | 300.96 | 309.79 | 305.75 | 296.61 | 143.79 | 119.23 | 129.33 | 130.78 | 149.19 |
| **SAS** | 376.44 | 362.65 | 375.76 | 371.62 | 373.47 | 232.43 | 252.54 | 230.45 | 238.47 | 207.03 |
| | **H [kg/kg]** | | | | | **Ps [HPa]** | | | | |
| | *GFDL* | *IPSL* | *MIROC* | *Ens.Mean* | *WFDEI* | *GFDL* | *IPSL* | *MIROC* | *Ens.Mean* | *WFDEI* |
| **NEU** | 0.0051 | 0.0048 | 0.0066 | 0.0055 | 0.0055 | 995.14 | 994.72 | 992.99 | 994.28 | 983.13 |
| **MED** | 0.0075 | 0.0075 | 0.0087 | 0.0079 | 0.0076 | 981.06 | 979.10 | 980.40 | 980.19 | 958.26 |
| **NEE** | 0.0042 | 0.0041 | 0.0054 | 0.0046 | 0.0045 | 998.58 | 997.13 | 995.35 | 997.02 | 994.48 |
| **NAS** | 0.0031 | 0.0036 | 0.0042 | 0.0037 | 0.0033 | 966.94 | 964.29 | 964.13 | 965.12 | 955.25 |
| **CAS** | 0.0044 | 0.0044 | 0.0057 | 0.0048 | 0.0055 | 900.50 | 896.25 | 899.36 | 898.70 | 893.06 |
| **TIB** | 0.0033 | 0.0034 | 0.0042 | 0.0036 | 0.0034 | 735.65 | 728.50 | 736.90 | 733.68 | 734.45 |
| **EAS** | 0.0090 | 0.0089 | 0.0108 | 0.0096 | 0.0078 | 974.67 | 969.55 | 973.25 | 972.49 | 947.43 |
| **SEA** | 0.0176 | 0.0178 | 0.0186 | 0.0180 | 0.0176 | 1000.13 | 1001.34 | 1003.18 | 1001.55 | 977.85 |
| **NAU** | 0.0121 | 0.0117 | 0.0140 | 0.0126 | 0.0096 | 991.65 | 994.78 | 994.03 | 993.49 | 978.92 |
| **SAU** | 0.0079 | 0.0068 | 0.0081 | 0.0076 | 0.0071 | 1004.23 | 1001.10 | 1002.27 | 1002.53 | 988.15 |
| **SAH** | 0.0061 | 0.0055 | 0.0068 | 0.0061 | 0.0061 | 965.67 | 965.58 | 966.70 | 965.98 | 955.18 |
| **WAF** | 0.0132 | 0.0123 | 0.0145 | 0.0133 | 0.0124 | 982.76 | 982.58 | 982.96 | 982.77 | 970.86 |
| **EAF** | 0.0113 | 0.0112 | 0.0130 | 0.0118 | 0.0122 | 939.81 | 936.28 | 940.58 | 938.89 | 928.97 |
| **EQF** | 0.0126 | 0.0135 | 0.0132 | 0.0131 | 0.0131 | 927.28 | 923.68 | 927.22 | 926.06 | 897.12 |
| **SQF** | 0.0134 | 0.0136 | 0.0144 | 0.0138 | 0.0123 | 964.04 | 963.95 | 964.50 | 964.16 | 924.14 |
| **SAF** | 0.0104 | 0.0094 | 0.0104 | 0.0101 | 0.0077 | 970.87 | 970.37 | 970.88 | 970.71 | 909.10 |
| **WNA** | 0.0059 | 0.0062 | 0.0074 | 0.0065 | 0.0051 | 908.11 | 909.20 | 907.96 | 908.42 | 867.44 |
| **CNA** | 0.0071 | 0.0067 | 0.0078 | 0.0072 | 0.0071 | 970.30 | 967.75 | 964.45 | 967.50 | 967.64 |
| **ENA** | 0.0092 | 0.0097 | 0.0113 | 0.0101 | 0.0068 | 1005.31 | 1003.65 | 1001.77 | 1003.58 | 986.35 |
| **CAM** | 0.0135 | 0.0136 | 0.0147 | 0.0140 | 0.0122 | 983.62 | 983.88 | 982.98 | 983.49 | 928.03 |
| **AMZ** | 0.0135 | 0.0140 | 0.0158 | 0.0144 | 0.0158 | 969.59 | 970.66 | 970.49 | 970.25 | 956.50 |
| **CSA** | 0.0100 | 0.0091 | 0.0096 | 0.0096 | 0.0095 | 976.00 | 975.62 | 973.88 | 975.17 | 935.84 |
| **SSA** | 0.0060 | 0.0047 | 0.0057 | 0.0055 | 0.0050 | 997.59 | 994.17 | 993.09 | 994.95 | 957.83 |
| **SAS** | 0.0134 | 0.0136 | 0.0152 | 0.0141 | 0.0132 | 965.75 | 965.46 | 965.67 | 965.63 | 932.59 |
| | **W [m/s]** | | | | | | | | | |
| | *GFDL* | *IPSL* | *MIROC* | *Ens.Mean* | *WFDEI* | | | | | |

| | | | | | |
|---|---|---|---|---|---|
| **NEU** | 5.50 | 4.47 | 4.10 | 4.69 | 3.64 |
| **MED** | 4.02 | 3.99 | 4.32 | 4.11 | 3.17 |
| **NEE** | 3.61 | 2.93 | 3.01 | 3.18 | 3.56 |
| **NAS** | 3.57 | 3.46 | 3.85 | 3.63 | 3.05 |
| **CAS** | 2.85 | 3.64 | 4.33 | 3.61 | 3.27 |
| **TIB** | 2.46 | 3.98 | 5.50 | 3.98 | 3.49 |
| **EAS** | 4.54 | 4.39 | 4.18 | 4.37 | 3.15 |
| **SEA** | 5.09 | 3.75 | 3.89 | 4.24 | 1.83 |
| **NAU** | 4.48 | 3.93 | 4.24 | 4.22 | 4.24 |
| **SAU** | 6.46 | 6.87 | 7.14 | 6.83 | 4.16 |
| **SAH** | 3.59 | 4.12 | 4.53 | 4.08 | 4.33 |
| **WAF** | 2.84 | 2.54 | 3.12 | 2.83 | 2.77 |
| **EAF** | 2.95 | 3.23 | 3.85 | 3.34 | 3.24 |
| **EQF** | 3.08 | 2.75 | 3.19 | 3.01 | 2.68 |
| **SQF** | 3.82 | 3.55 | 4.01 | 3.79 | 2.49 |
| **SAF** | 5.15 | 5.40 | 5.78 | 5.44 | 3.79 |
| **WNA** | 3.88 | 3.50 | 4.78 | 4.05 | 3.06 |
| **CNA** | 3.29 | 3.28 | 3.34 | 3.30 | 3.90 |
| **ENA** | 5.22 | 4.72 | 4.46 | 4.80 | 2.86 |
| **CAM** | 4.48 | 3.89 | 4.55 | 4.31 | 2.50 |
| **AMZ** | 2.91 | 2.73 | 2.10 | 2.58 | 1.71 |
| **CSA** | 4.68 | 4.83 | 5.11 | 4.87 | 3.24 |
| **SSA** | 7.94 | 7.90 | 8.54 | 8.12 | 5.14 |
| **SAS** | 4.31 | 3.56 | 3.13 | 3.67 | 2.49 |

*RC3. Structural, the paper misses a clear separation between "Results" and "Discussion". For example, section "The model evaluation. . ." at page 11 reads for me like a discussion (finding out reasons for performance of the model). Please move to discussion part. Another example is Page 13 section starting with "First.." – the authors itself write that they discuss. Please avoid that in a results section. Similar difficulties I have with P15, section starting with "The variation..". You could also consider to have a joint "Results and discussion".*

AC3. The reviewer's concerns on the structures issues are valid. Thus, following the reviewer's suggestion, "Results and Discussion" is a joint version in the revised manuscript. This option was preferred because reviewer #2 suggested that our discussion on runoff sensitivities to specific humidity should take a more prevalent role in the manuscript and that supplementary figures regarding this section should be included in the manuscript.

*Minor*

*RC4. P5, l16: "been used in the BCIP": Could you please write some essentials of the intercomparison results? This is especially of importance, as it seems that the method is only applied in studies by the authors of this manuscript.*

AC4. Beyond the aforementioned studies of the authors that make use of the bias correction methodology, MSBC has also been used in the framework of ECLISE FP7 and HELIX FP7 projects. In the latter, the methodology is used to adjust biases in a range of climate variables such as radiation (rlds, rsds), temperature (tas, tasmax, tasmin), wind, precipitation and specific humidity.

The BCIP project aims to address a number of topics and bias adjustment related gaps in use of climate information. To this time, the comparison has been narrowed to precipitation, mean temperature, maximum and minimum temperature (pr, tas, tasmax, tasmin). As a first stage of comparison, methods are assessed for their ability to reproduce mean values of the observational dataset, as well as lower and upper percentiles ($1^{st}$, $5^{th}$, $95^{th}$ and $99^{th}$ percentiles) and the 20-year return period, on data from two RCMs. While the project is still ongoing, a first set of results can be found in Nikulin et al. (2015). Relevant presentation slides, presenting the remaining bias in mean precipitation for the two RCMs, can be accessed with the link below.

http://www.meteo.unican.es/files/posters/20150415_EGU2015_BCIP_GN.pdf

The MSBC methodology is found to perform well in all metrics, comparing to the other methodologies, ranking it high in performance. Respective comparison for the remaining bias in mean and the high/ low percentiles of the four variables and two RCMs are available upon demand. A representative comparison of the remaining bias in temperature and precipitation at mean and $99^{th}$ percentile is shown for DJF and JJA periods in the figure below.

The following part was added to the revised manuscript (P5,L31 in Section 2.4 Bias correction method) to summarize the different uses and performance of the employed bias correction method: "The methodology has already been used in in the framework of ECLISE FP7 (265240) and HELIX FP7 (603864) projects and in a number of climate change impact studies (Grillakis et al., 2016; Papadimitriou et al., 2016). Additionally MSBC has participated in the Bias Correction Intercomparison Project (BCIP) (Nikulin et al., 2015), where it was found to compare well to the other methodologies and was ranked high in performance."

[Figure]

**Figure R 1. Comparison of the remaining bias in temperature and precipitation at mean and 99th percentile for DJF and JJA periods, using different bias correction methods**

*RC5. P5, l28: could you somehow describe "edge segments" e.g. by a percentile? Otherwise it reads a bit vague.*

AC5. As it is described in the methodology section, the bias adjustment method partitions the data CDF into discrete segments and applies a quantile mapping correction to each segment, achieving a better fitted transfer function. The optimal number of the segments is estimated by Schwarz Bayesian Information Criterion to balance between complexity and performance. Additionally the upper and lower CDF segments (Segments 1st and 5th in Figure R1, for an example of an optimum 5 segments) are explicitly corrected using the average difference between the reference period of the raw model data and the observations (ΔT). As the number of segments is not predefined, there are not fixed percentile values to define the "edge" segments. The description of the methodology in the revised manuscript has been enriched. However, technical details are not described in depth as they are beyond the main focus of the study. A detailed assessment of the methodology along with a full technical description of the method has just been submitted for consideration in Earth System Dynamics-Discussion (http://www.earth-syst-dynam-discuss.net/esd-2017-53/) in Grillakis et al. (2017).

[Figure]

**Figure R 2. Edge segments of the CDFs in a theoretical example of 5 segments and the ΔT correction applied to them (borrowed and modified from (Grillakis et al., 2017)).**

*RC6. P8, l7: what is meant by median value? Of each grid cell in a specific region? Please specify.*

AC6. Median values of change (median|ΔRF|) and sensitivity (median|S|) are derived by considering the values of all land grid boxes and for all the experiments (apart from temperature). This way, |ΔRF| and |S| of each grid box and each experiment are compared against the same value.

The clarification is added to the revised manuscript (P8, L15-20 of the revised manuscript).

*RC7. P11 section starting with "The shown persistent. . .": Although interesting, it is a bit vague and could be supported by citing common papers (e.g. Coxon et al. (2015) for discharge uncertainty). Please either move the last sentence "We believe. . ." to discussion and discuss it properly or just delete it. It is too speculating without providing reasons for this statement.*

AC7. Following the reviewer's indications, relevant references were added to support this section and the last sentence was deleted. The new section which can be found in P12 of the revised manuscript is:

"The shown persistent departure from the mean climatology of discharge could include three types of errors. The first is the error stemming from the insufficient description of the runoff processes by the land surface model and from the routing algorithm (Blyth et al., 2011). The second type of error is a result of errors in the forcing datasets (either observational or GCM output) with regards to depicting the real climatic drivers (Elsner et al., 2014; Mizukami et al., 2014). A third possible error comes from the comparison of naturalized discharge of the simulations with measured discharge due to influences like abstractions and dams regulating the natural river flow (Müller Schmied et al., 2014). An extra error component, which is not considered here, could result from the uncertainty in discharge measurements (Coxon et al., 2015)."

*RC8. P12, l12: please check the statement, that global LSMs are calibrated (the sentence reads so).*

AC8. In this sentence, we did not want to state that LSMs are calibrated, but rather highlight that tuning of their parameters would be a very complicated and time consuming task, which is done for specific applications and locations. Thus, even if the model parameters are tuned for a specific application and location, there are still going to exist regions of lower performance at the global scale. However, because the statement about LSM calibration is confusing and does not help the message we are trying to communicate in this part of the paper (that the global nature of the model does not allow top performance for all the regions), this part of the sentence was deleted in the revised manuscript.

*RC9. P16, l15: I cannot see terrain elevation in Fig. 11, so the statement cannot be made (strictly speaking).*

AC9. The statement was replaced with the following sentence in the revised manuscript (P16,L31): "The highly affected areas mainly correspond to regions with high mountain ranges."

*RC10. Please go carefully through the reference list. A quick look on it shows a lot of inconsistencies. For example: first reference – Journal name is missing – and why are you citing the discussion paper and not the final version? Check consistency of Giorgi and Bi. Check if everywhere a doi is provided, check if upper/lower case is consistent, Hattermann et al 2016 is published since a while (please update citation), and what does "Submitted in this special issue" should mean? N/a in Maraun 2012? What are "and Ohters" in Nikulin et al?, Journal / doi for Oki and Sud? I did not check if all references are listed in the reference list / in the manuscript.*

AC10. In the revised manuscript, the reference list has been thoroughly checked and the inconsistencies have been corrected.

*RC11. Figure 7: I am a bit sceptic to consider a NSE > 0 as "good". In many studies, this is the case of e.g. > 0.5 or 0.7. It is absolutely not necessary to provide color hues to indicate how good or bad a model performs, it hinders (me) for an objective look at the table (in fact, it is a table with colored cells). Please convert to a real table for more clarity.*

AC11. The value of zero for NSE is set as an arbitrary limit to characterize the model behavior as "acceptable", rather than "good". In our case, the lower value of NSE for the run forced with WFDEI data is 0.24. The reviewer's considerations are valid regarding basin scale hydrological applications, especially when these are conducted with basin scale, calibrated models. However, in the case of global modelling the evaluation metrics can have looser thresholds, as it is unrealistic that the model will have such a good performance (NSE>0.5 or 0.7) for many different basins, run with the same model configuration which is "tuned" on land processes rather than strictly on runoff representation-. Most global scale hydrological evaluations avoid using the NSE index and employ metrics such as the RMSE and the correlation coefficient (e.g. Blyth et al., 2011; Hattermann et al., 2017). In the studies that the NSE metric is used, it is not unusual to encounter negative values of this evaluation index (e.g. MacKellar et al., 2013; Zulkafli et al., 2013). In our evaluation, we employ three different metrics, in addition to the visual inspection of the annual cycles, in order to have a multi-faceted assessment of model performance.

Following the reviewer's indication, the colored table presented in figure form was converted to a real table, added to the tables of the manuscript (as Table 3).

*RC12. Figure 11: Please consider other colors to distinguish ECII and ECI for better visualization.*

AC12. The authors considered the reviewer's suggestion but concluded that the color hues of ECI and ECII are distinguishable and do not necessarily need alterations in order to be understood by the reader. Moreover, the color hues were carefully selected to reflect the relationship between the change and sensitivity. In more detail, a warm tone was selected for the high change category and a grey tone for the

low change category. Then, the saturation of each tone increases for the high sensitivity category (resulting to dark orange for ECI and dark grey for ECIII).

*Supplement:*

*RC13. Table S1 is not referred from the main manuscript, please also provide station name. Fig. S3 – what does the red color mean? Please indicate in figure caption. Figs S4-S10 are not mentioned in the main manuscript. To my knowledge, a supplement should support the main paper, and that are interesting figures, but without referencing it in the main paper, they are unconnected and lost with just the figure caption.*

AC13. The station name was added to the information provided in Table S1. In Fig. S3, the red color indicates the selected focus regions, which are also included in the main body of the manuscript. A relevant explanation is added to the legend of the figure. Following the reviewer's indications, all the tables and figures presented in the Supplement are referenced in the manuscript.

*Technical*

*RC14. P1, l26: please be consistent: either Global Hydrological Model or global hydrological model, not global Hydrological Model.*

AC14. The inconsistency in the text is revised to "Global Hydrological Model".

*RC15. P3, l29: check if Penman (1948) is the correct citation for Penman-Monteith approach (isn't it Monteith 1965?)*

AC15. The reviewer is correct. The reference has been changed in the revised manuscript.

*RC16. P5, l 4 (the two sentence starting with "The WFDEI". . .): I feel this information is not required for the manuscript. Consider to shorten it.*

AC16. The sentence is shortened in the revised manuscript.

*RC17. P6, l31: please be as specific as possible in naming the variables. Is it net shortwave radiation, or downward shortwave / longwave (which I am sure is meant?)*

AC 17. All the radiation components refer to downward radiation. Relevant clarifications were added to the manuscript.

*RC18. P12, l26: please insert a blank between number and unit. Same at l30 (5mm)*

AC18. Corrections have been made in the revised manuscript.

*RC19. P16, l6: to which section are you referring to?*

AC19. To text was referring to Section 2.8 (Categorization of individual bias effects). The correct reference is added in the revised manuscript.

We would like to thank Anonymous Referee #2 for providing us with constructive comments on the submitted manuscript. We believe that the input that will help improve the manuscript significantly. The reviewer's comments (shown in italics) have been addressed point-by-point.

Anonymous Referee #2

*General comments:*

*This paper examines the impact of bias correction on global uncoupled land-surface model (LSM) simulations using GCM data as inputs. They step through an observation-based simulation, and two GCM simulations using raw GCM inputs and a bias correction simulation that corrects all seven variables required to run an LSM that solves the surface energy balance. They find the bias correction scheme removes nearly all bias from all seven variables and the subsequent LSM simulations closely match the observation-based simulation.*

*They then step through six simulations where they remove bias correction from one variable at a time (e.g. precipitation or temperature) to examine the impact of bias correction for that specific variable. Findings highlight that precipitation and temperature are the most important variables to bias correct, as expected. However, radiation is also found to have high sensitivity and should be corrected as well. Surface pressure and wind speed generally have little impact and can almost always be neglected. I appreciate the literature review to place the sensitivities found here in the context of other studies.*

*Extra time is spent examining humidity, as extreme sensitivity across higher latitudes in the northern hemisphere is found in the LSM (JULES in this paper). They attribute the extreme sensitivity to reduced ET in high humidity environments, and direct condensation and deposition of water vapor into the snowpack in these regions.*

*Overall, this paper is logically organized and generally easy to follow. I think it will be ready for publication in HESS after the authors address my comments.*

*Comments:*

*RC1. The specific humidity discussion is much needed to explain the extreme sensitivities in the JULES model. I believe it should be expanded in the main text, with the figures in the supplemental material moved to the paper. Additionally, examination of relative humidity for supersaturated conditions should be done for an example grid cell or the entire Northern European region.*

*Figure S.4 and the discussion in Section 4.4 imply that the raw GCM runs are too humid and have supersaturated conditions in high latitudes in the northern Hemisphere. I'm a bit surprised by this, thus it is worth checking in more detail by looking at relative humidity over a larger spatial region in the raw GCM output.*

*What GCM data are used to force JULES? That is a key detail I cannot find in the paper. Was it the lowest sigma level of the GCM, or the 10-m/2-m diagnosed variables from the GCM? It is possible (although maybe not likely) that the issue could stem from the use of JULES while the GCMs use a different LSM and that mismatch could be at the root of these results.*

*Finally, some detailed examination of how JULES treats vegetation and snowpack versus the LSMs in the GCMs may also give some insight into this issue. These regions are primarily boreal forest, which are sometimes difficult areas to model in winter, particularly surface fluxes into/out of the snowpack (e.g. Chen et al. 2014). Is it possible to distinguish between condensation and deposition in the JULES output? That may give additional insight into JULES behavior.*

AC1. Figures S4, S5 and S6 of the Supplementary material were combined in one Figure showing 3 plots of latitudinal means and were transferred to the main manuscript, after the reviewer's suggestion (Figure 11, shown below). We would prefer to leave the other two relevant figures (showing the seasonality of snowmass and the difference in long term means of water fluxes respectively) as Supplementary figures, in order to keep a proper number of figures (already 11) in the main manuscript.

[Figure]

**Figure 1. a. Latitudinal means of raw and bias corrected specific humidity [g/kg], b. Latitudinal means of JULES' runoff forced with raw and bias corrected specific humidity [mm/day], c. Percent differences of the latitudinal means in a (H) and b (RF). The latitudinal means are calculated from the 1981-2010 period.**

The reviewer suggested an investigation of relative humidity for supersaturated conditions. From the input specific humidity H, we estimated the respective relative humidity (this transformation also requires temperature T and surface pressure Ps as input to the Clausius-Clapeyron equation). Then we calculated the fraction of time (based on a daily timestep) that supersaturated conditions occur, for the historical period 1981-2010. The estimation was performed for a) the WFDEI H, T, Ps, b) the raw H, T, Ps, c) the bias corrected H, T, Ps and d) for a combination of data corresponding to the NobcH run (raw H combined with bias corrected T and Ps). The results are presented in the following figure, which has been added to the Supplement of the paper:

[Figure]

**Figure S 1. Fraction of time under supersaturated air conditions (Relative humidity>100%), calculated from specific humidity H, temperature T and surface pressure Ps for: a. WFDEI data, b. Raw GCM data, c. BC GCM data and d. data corresponding to NobcH (raw H, BC T and BC Ps). Calculation of relative humidity uses the Clausius-Clapeyron equation. Fraction of time refers to the historical period 1981-2010.**

Firstly, this analysis reveals that supersaturated conditions are present in all three sets of datasets (WFDEI, Raw and BC). Is also worth noting that supersaturation was observed both in cases with daily temperature above and below 0ºC. Secondly, the NobcH calculation has a pattern of supersaturated conditions over the northern latitudes, corresponding to the high sensitivity regions discussed in the relevant section of the manuscript (Section 3.8.4 of the revised manuscript). Thus, the high runoff sensitivity over the high latitude regions is not a result of supersaturated conditions in the raw GCM H and it rather stems from: 1)

raw GCM H being higher than BC H and 2) the calculation of relative humidity within JULES, done by combining raw GCM H with bias corrected T and Ps.

This inconsistency strengthens the argument for the need of bias correction of more forcing variables -in addition to P and T. Specific humidity is a variable that is often left uncorrected, a practice that could possibly result to runoff overestimations in the northern latitudes based on our findings, in cases that hydrological models which account for deposition and condensation are used.

A discussion on these findings is added to the relevant section in the revised manuscript (P19, section starting with: "A comparison of super-saturated air conditions…").

The GCM data used to run JULES are described as "near surface" fluxes, which typically refers to 2 m above ground. On the consideration of the reviewer about the issues possibly stemming from the use of JULES while the GCMs use other LSMs, we think that this as an unlikely implication. JULES, as most LSMs, is designed to run both online with GCMs and offline, using GCM output data as forcing. Moreover, this would question the widely used practice of the scientific modelling community of forcing GHMs and LSMs with GCM output data in order to perform impact assessments.

The supersaturated conditions revealed in Figure S6 result from the interaction of atmospheric only variables (specific humidity, temperature and surface pressure). The JULES model performs a respective calculation prior to calculating the water and energy fluxes. In the presence of supersaturated conditions in the model, water vapor transitions to liquid (condensation) or ice (deposition), depending on temperature. As supersaturated conditions here result from the combination of raw H with bias corrected T and Ps, we believe that a comparison of vegetation and snow models used in the LSM of the GCM, although interesting, cannot add relevant information to our analysis. The reviewer's point on the difficulty for modelling the snow dominated boreal forest regions is an important issue, which will particularly affect the sensitivity of GHMs and LSMs for these regions. In our single model study, we cannot assess possible sensitivities of other models. However, our study highlights the importance for special focus on the snow dominated boreal forest in future studies that will assess sensitivities of multiple models, due to the complex interactions between vegetation, snow and energy and radiation fluxes in the aforementioned regions.

*RC2. Discussion in sections 3.1 and 3.2 can be shortened to make room for the expanded discussion of the specific humidity biases. These add little value to the paper in my opinion.*

AC2. The authors tried to shorten these sections but it was not easy to leave out much of the presented content. The discussion of the results in sections 3.1 and 3.2 was requested by reviewer#1 during our previous submission of the manuscript, so we cannot substantially shorten these parts. Moreover, we believe that the results are discussed quite briefly and it is due to the number of examined variables that the sections increase in length.

*RC3. The bias correction procedure needs more detail discussed. The authors mention the issues of keeping variables physically consistent when performing bias correction across many variables. However, they do not discuss the basic checks that are still necessary and should be performed when bias correcting many variables. For example, limiting humidity to prevent supersaturation conditions in the bias corrected fields when correcting temperature and specific humidity.*

AC3. A number of essential parameter checks were performed along with the bias adjustment of the variables, such as prevention of unrealistic values (e.g. negative values to positively constrained variables) and the avoidance of extreme values well beyond or below the historical record of WFDEI. As the bias correction was limited to the historical period, this was a trivial task, as transfer functions were not extrapolated to a projection period. The bias adjustment for each calendar month aided the physical consistency of the bias adjusted variables, as it adjusts the seasonality in a coherent way according to the observational dataset that is used.

The issue of humidity supersaturation was also investigated in the bias correction stage. As it is discussed and shown in previous comment, supersaturation is a rare occurring state, which however exists in both raw climate model data and the WFDEI observational dataset, many of the times in cases where mean daily temperature exceeds 0°C. Masaki et al., (2015) also report supersaturation even at temperatures over 0°C, mainly in cases of raw GCM data. Additionally, WFDEI meteorological forcing, which is explicitly developed for hydrological purposes, has been included to the ISIMIP2b (Frieler et al., 2016) protocol without any notation about humidity treatment prior to its use. Considering these, it was decided to avoid treating the supersaturation cases in the WFDEI and the raw and bias corrected datasets, as it is inherent in data. Nonetheless, as discussed in the answer of the first comment of the reviewer, runoff sensitivity to specific humidity biases is driven by the combination of bias adjusted temperature and raw specific humidity.

References:

Frieler, K., Betts, R., Burke, E., Ciais, P., Denvil, S., Deryng, D., Ebi, K., Eddy, T., Emanuel, K., Elliott, J., Galbraith, E., Gosling, S. N., Halladay, K., Hattermann, F., Hickler, T., Hinkel, J., Huber, V., Jones, C., Krysanova, V., Lange, S., Lotze, H. K., Lotze-Campen, H., Mengel, M., Mouratiadou, I., Müller Schmied, H., Ostberg, S., Piontek, F., Popp, A., Reyer, C. P. O., Schewe, J., Stevanovic, M., Suzuki, T., Thonicke, K., Tian, H., Tittensor, D. P., Vautard, R., van Vliet, M., Warszawski, L. and Zhao, F.: Assessing the impacts of 1.5 °C global warming - simulation protocol of the Inter-Sectoral Impact Model Intercomparison Project (ISIMIP2b), Geosci. Model Dev. Discuss., 1–59, doi:10.5194/gmd-2016-229, 2016.

[revised manuscript text omitted]